# Transition metal dichalcogenides bilayer single crystals by reverse-flow chemical vapor epitaxy

Xiumei Zhang[1,2], Haiyan Nan[1], Shaoqing Xiao [1], Xi Wan[1], Xiaofeng Gu[1], Aijun Du[3],
Zhenhua Ni[4] & Kostya (Ken) Ostrikov[3,5]

Epitaxial growth of atomically thin two-dimensional crystals such as transition metal dichalcogenides remains challenging, especially for producing large-size transition metal dichalcogenides bilayer crystals featuring high density of states, carrier mobility and stability at room temperature. Here we achieve in epitaxial growth of the second monolayer from the first monolayer by reverse-flow chemical vapor epitaxy and produce high-quality, large-size transition metal dichalcogenides bilayer crystals with high yield, control, and reliability. Customized temperature profiles and reverse gas flow help activate the first layer without introducing new nucleation centers leading to near-defect-free epitaxial growth of the second layer from the existing nucleation centers. A series of bilayer crystals including $MoS_2$ and $WS_2$, ternary $Mo_{1-x}W_xS_2$ and quaternary $Mo_{1-x}W_xS_{2(1-y)}Se_{2y}$ are synthesized with variable structural configurations and tunable electronic and optical properties. The robust, potentially universal approach for the synthesis of large-size transition metal dichalcogenides bilayer single crystals is highly-promising for fundamental studies and technological applications.

---

[1] Engineering Research Center of IoT Technology Applications (Ministry of Education), Department of Electronic Engineering, Jiangnan University, 214122 Wuxi, China. [2] School of Science, Jiangnan University, 214122 Wuxi, China. [3] Institute for Future Environments and School of Chemistry, Physics and Mechanical Engineering, Queensland University of Technology, Brisbane QLD 4000, Australia. [4] Department of Physics and Key Laboratory of MEMS of the Ministry of Education, Southeast University, 211189 Nanjing, China. [5] CSIRO-QUT Joint Sustainable Processes and Devices Laboratory, Commonwealth Scientific and Industrial Research Organization, P.O. Box 218 , Lindfield NSW 2070, Australia. These authors contributed equally: Xiumei Zhang, Haiyan Nan. Correspondence and requests for materials should be addressed to S.X. (email: larring0078@hotmail.com) or to X.G. (email: xgu@jiangnan.edu.cn)

Atomically thin two-dimensional (2D) crystals of transition metal dichalcogenides (TMDs) have been studied with great attention in view of their unique properties and potential applications in electronic and optoelectronic devices[1–4]. Monolayer TMD crystals[5–10] have small state densities which may limit their practical device applications. Few-layer, especially bilayer (2L) TMD crystals, are particularly attractive for applications in thin film transistors[11], logic devices[12], and sensors[13] owing to their high density of states, carrier mobility and stability at room temperature[11,14]. Moreover, bilayer materials are important basic structures for studying atomic interaction between single layers. Despite considerable efforts and limited success to date, reliable synthesis of large-size high-quality TMD bilayer single crystals remains a significant challenge, in part due to major issues in controllable epitaxial growth of the second monolayer from the first monolayer.

Bilayer or few-layer TMD materials used for applications[11–13] and fundamental studies[14–20] are mainly derived through micromechanical exfoliation, limited by low yield, poor controllability in layer number and size. Post-treatment using laser[21], plasma[22,23], patterning method[24], and thermal annealing[25] also produces few-layer, bilayer, and monolayer flakes but show same constrains. Contrary to these top-down methods, the bottom-up chemical vapor deposition (CVD) approach is highly regarded as one of the most promising ways to produce TMD multilayers[26–29]. However, the as-produced TMD structures are typically polycrystalline with the many randomly oriented domains and domain boundaries, which inevitably diminish their device performance[29–33]. Other common issues include co-existence of areas with monolayer, bilayer, trilayer and few-layer structures, small crystal size (typically in the 10–30 μm range), and low process yield[26,34].

These issues are particularly daunting for bilayer TMD crystals because CVD growth requires at least two stages with different growing temperatures, adjusted to enable high-order stacking of single layers[26,35–38]. Unfortunately, TMD monolayer crystals are usually too delicate to survive multiple sequential stages required for homogeneous epitaxy of the second monolayer and relaxation of as-formed bilayer crystals. Poor control of chemical vapor during the temperature swing stage between the growth stages often leads to uncontrolled and undesired nucleation centers on the first monolayer. To achieve large size and high quality TMD bilayer single crystals, it is essential to ensure homogeneous epitaxy of the second monolayer from activated nucleation centers of the first monolayer while suppressing new nucleation centers.

Here we propose a concept of a potentially universal sequential two-stage thermal CVD process, wherein a reverse hydrogen flow is introduced and the process temperature is adjusted during the growth swing stage for the first and second layer, to simultaneously satisfy both of the above requirements. The reverse hydrogen flow is demonstrated to be beneficial in both reducing the undesired nucleation centers and promoting homogeneous epitaxy of the second monolayer from the activated nucleation centers of the first monolayer. This approach is used to synthesize high-quality TMD bilayer crystals (e.g., $MoS_2$) with high yield, large size, and high controllability. The as-grown $MoS_2$ samples exhibit uniform and dominant bilayer domains with AA and AB stacking structures. The maximum lateral size of such bilayer domains can reach 300 μm, well above typically achievable otherwise. A series of advanced characterization tools including optical microscopy, Raman, photoluminescence (PL), atomic force microscopy, and electrical transport measurements are used to evaluate the quality of the as-grown $MoS_2$ bilayer samples. Back-gated field-effect transistor (FET) devices based on the as-grown AA stacking $MoS_2$ bilayer samples show better electronic performances compared to monolayer TMD-based devices. Our reverse-flow chemical vapor epitaxy is also applied for the synthesis of various high-quality TMD bilayer crystals, including $WS_2$, ternary $Mo_{1-x}W_xS_2$, and even quaternary $Mo_{1-x}W_xS_{2(1-y)}Se_{2y}$ alloys.

## Results

**Synthesis and characterization of bilayer $MoS_2$.** Bilayer $MoS_2$ grains were grown by the modified sequential two-stage thermal CVD process. The experiment setup and the temperature program are shown in Fig. 1a and Supplementary Figure 1, respectively. Both sides of the CVD tube are equipped with gas inlet and outlet. The direction of gas flow can be switched by simultaneously turning on gas valves 1 and 4 (or gas valves 2 and 3) and turning off gas valves 2 and 3 (or gas valves 1 and 4). The growth process is intentionally divided into two sequential growth stages to enable high-order stacking of single layers: A-B stage stands for the growth of first layer, C-D stage represents the growth of second layer, while B-C stage corresponds to the growth swing stage for the first and second layers. The lower growth temperature (700 °C) of A-B stage is enough for the growth of the first monolayer crystals, while the higher growing temperature (750 or 800 °C) is beneficial for the vertically epitaxial growth of the second monolayer crystals but is against to the continued laterally epitaxial growth at the edge of the first monolayer crystals[26,35–38]. In fact, only $MoS_2$ monolayer grains can be obtained (Supplementary Figure 2) when the growing temperature was maintained at 700 °C throughout the whole growing process from A-B to C-D stage. This could be well understood since the laterally epitaxial growth dominated such growing process. In a typical sequential CVD growth process with the forward gas carrier flow direction, the uncontrolled nucleation during the temperature rising process between sequential growth stages leads to many triangle monolayer crystals (second layer) randomly growing on the surface of the first monolayer crystals, as shown in Supplementary Figure 3. To solve this problem, a reverse $N_2/H_2$ flow from the substrate to the source was introduced during the temperature swing stage (B-C stage). The synthesis details and challenges in bilayer growth are described in Supplementary Note I. At times, the adjacent bilayer $MoS_2$ domains may meet each other and merge together to form a continuous bilayer film as shown in Supplementary Figure 4. The grain boundaries can be clearly observed by naked eyes, and this situation is similar to the case of monolayer growth as reported[39].

The strikingly positive effect of such reverse $N_2/H_2$ flow during the temperature swing stage can be explained as follows. First, the reverse carrier gas flow can prevent unintended supply of chemical vapor source to eliminate the generation of new nucleation centers on the growth substrate and the as-grown first layer during the temperature rising process[40]. Zhang et al.[8] employed the same idea to prevent uncontrolled homogeneous nucleation and thus enabled highly robust epitaxial growth of diverse heterostructures, multi-heterostructures, and superlattices from 2D atomic crystals. Second, the hydrogen flow could saturate the dangling bonds on the edge and at the surface of the as-grown first $MoS_2$ monolayer crystals, thus blocking the laterally epitaxial growth as reported by Jia et al.[41]. The surface energy of the edge-terminated structure is considerably higher compared to that of the as-grown flat basal-plane structure[42,43]. Consequently, the second monolayer is more likely to deposit on the as-grown monolayer surface. At the same time, $H_2$ can slightly etch away emerging nucleation points on the growth substrate and thereby reduce the wettability of the growth substrate[44]. As such, the source vapor during the C-D stage can be easier to transfer through the substrate surface and have more

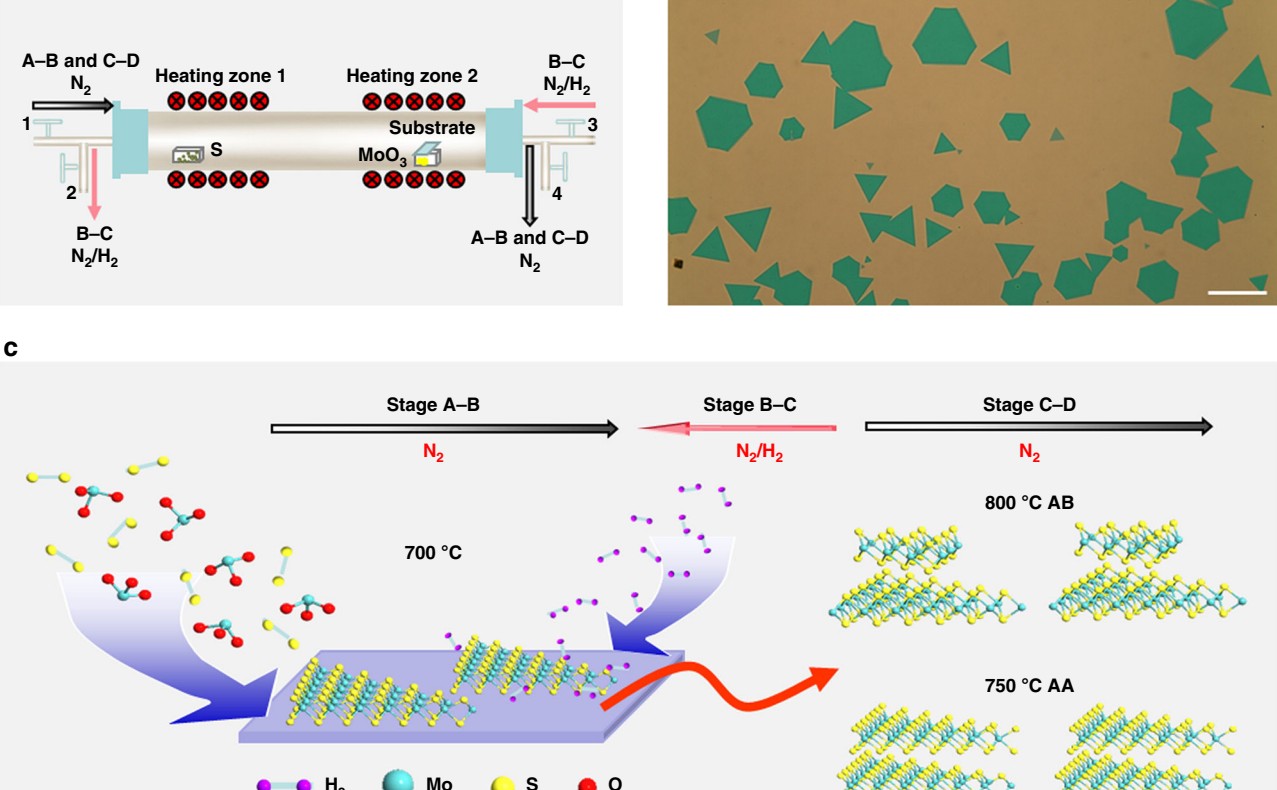

**Fig. 1** Reverse-flow chemical vapor epitaxy process for bilayer $MoS_2$. **a** Experimental setup of the modified sequential two-stage thermal CVD process: A-B stage stands for the growth of first layer, C-D stage represents the growth of second layer, while B-C stage corresponds to the growth swing stage for the first and second layers. A reverse $N_2/H_2$ flow from the substrate to the source was introduced during the temperature swing stage (B-C stage). **b** A representative optical image of the as-grown bilayer $MoS_2$ crystal grains. The scale bar is 100 μm. **c** Schematic diagram of the reverse-flow chemical vapor epitaxy process for bilayer $MoS_2$. Different growing temperatures at C-D stage can result in bilayer $MoS_2$ crystals with different stacking structures: 750 °C for AA stacking bilayer crystals and 800 °C for AB stacking ones

possibility to reach the surface of the as-grown first monolayer. With a suitable carrier gas flow rate, the source vapor can have enough kinetic energy to reach the surface center of the as-grown first monolayer where there is an initial nucleation to trigger the growth of the second layer[45]. Therefore, the second monolayer crystals prefer to grow epitaxially and homogenously on the activated nucleation centers of the first monolayer, finally promoting the growth of $MoS_2$ bilayer crystals, as shown in Fig. 1b.

From Fig. 1b, one can observe that large-area bilayer $MoS_2$ crystals with triangle and hexagon shapes are dominating over the substrate surface. Also, the optical contrasts of these bilayer crystals are not only quite uniform across the whole substrate surface but also deeper than that of the monolayer $MoS_2$ crystals as shown in Supplementary Figure 2. Furthermore, the different growing temperature at C-D stage can result in bilayer $MoS_2$ crystals with different stacking structures, and the results are presented in Supplementary Figure 5. The growth temperature difference between the two adjacent monolayers together with the reverse flow determines the stacking order. 750 °C is suitable for AA stacking $MoS_2$ bilayer crystals corresponding to 3R like phase in which the crystal orientation of the upper layer is consistent with that of the bottom layer[37], while 800 °C is optimal for AB stacking ones namely 2H phase in which the crystal orientation of the upper layer is opposite to that of the bottom layer[37]. The schematic diagram of the reverse-flow chemical vapor epitaxy process for bilayer $MoS_2$ crystals is displayed in Fig. 1c.

Figure 2a shows the optical microscope image of the as-grown complete AA stacking bilayer $MoS_2$ samples with the maximum lateral size of even up to 300 μm. The representative AFM image in Fig. 2b clearly reveal that the thickness of such crystals is about 1.4 nm, confirming the bilayer $MoS_2$ film characteristic[6]. Interestingly, the structure of the as-grown AA stacking bilayer $MoS_2$ samples displays a close relationship with the growth time of C-D stage (for the second layer growth), as presented in the optical images of the as-grown AA stacking bilayer $MoS_2$ samples under different growth time of C-D stage (Supplementary Figure 6). When the growth time of C-D stage was shortened to <10 min (like 5 min), the epitaxial growth of the second monolayer is insufficient to fully cover the surface of the first monolayer so that the incomplete AA stacking bilayer $MoS_2$ grains with distinct steps can be achieved as shown in Fig. 2c and Supplementary Figure 6a. The corresponding AFM image in Fig. 2d clearly shows the step between the substrate surface and the first monolayer, as well as that between the first monolayer and the second monolayer, both of which are characteristic of $MoS_2$ monolayer[46]. However, when the growth time was kept for about 10 min, both complete and incomplete AA stacking samples can coexist on the substrate surface as presented in Supplementary Figure 6b. When the growth time of C-D stage was kept for more than 12 min (like 15 and 20 min), the complete AA stacking bilayer samples without any steps can be predominately synthesized on the whole substrate surface as observed in Fig. 2a, Supplementary Figure 6c, d.

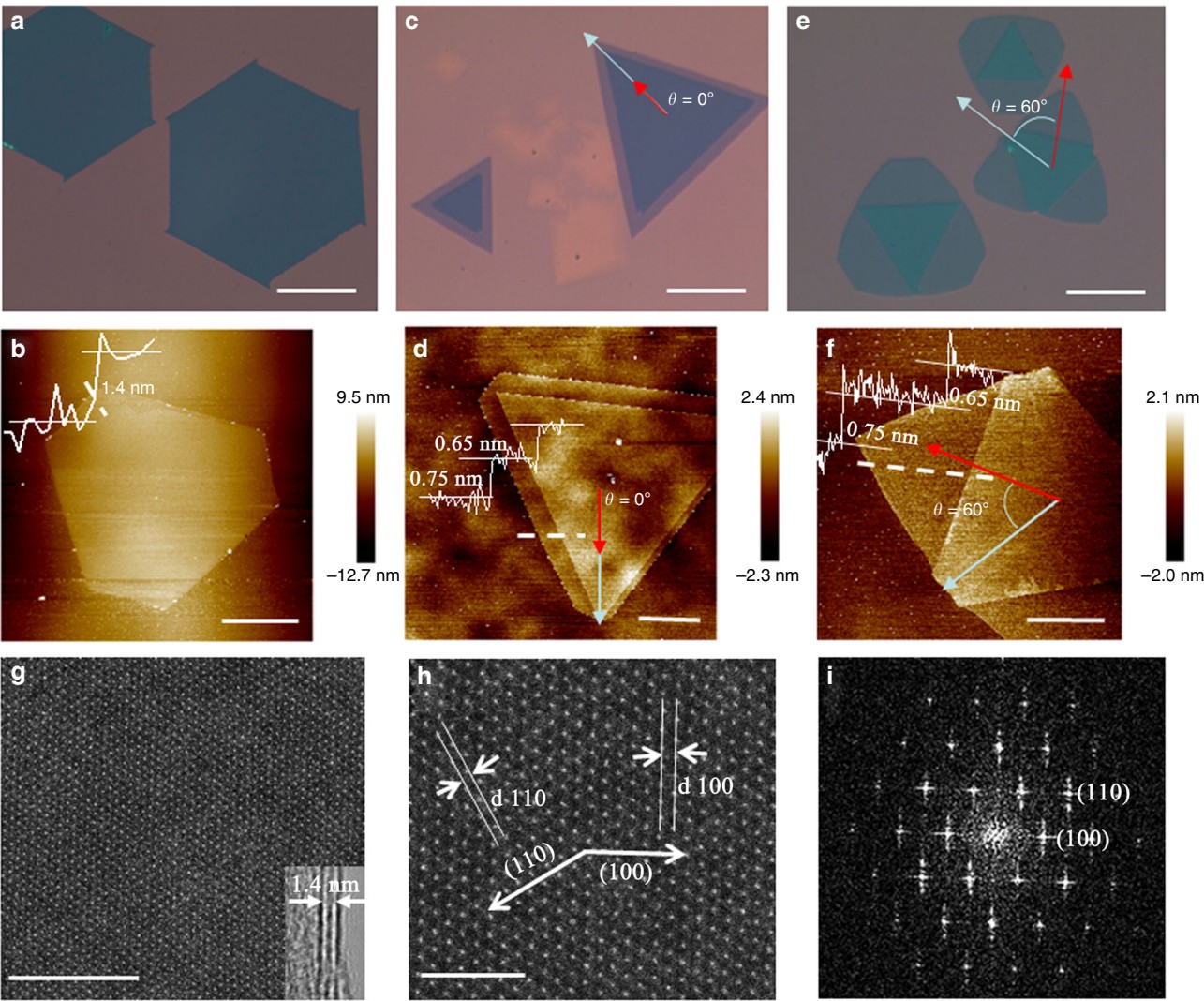

**Fig. 2** Characterization of the bilayer $MoS_2$ crystals. **a** A representative optical microscope image and **b** a representative AFM surface morphology image of the as-grown AA stacking bilayer $MoS_2$ samples with the maximum lateral size of even up to 300 μm. **c** A representative optical image and **d** a representative AFM surface morphology picture of the AA stacking bilayer $MoS_2$ grains with distinct steps, where the twist angle from the relative rotation of the two vertically stacked triangles $\theta = 0°$ indicate the AA stacking order[37,62,63]. **e** A typical optical image and **f** a typical AFM surface morphology image of the as-grown AB stacking bilayer $MoS_2$ samples, where the twist angle from the relative rotation of the two vertically stacked triangles $\theta = 60°$ indicate the AB stacking order[37,62,63]. Planar TEM images of the AA stacking bilayer $MoS_2$ crystals: **g** low resolution and **h** high resolution as well as **i** the FFT pattern. The inset in **g** shows the folded edge of $MoS_2$ bilayer films. Scale bars, 100 μm in **a**, **c**, and **e**, 20 μm in **b**, 2 μm in **d**, 5 μm in **f**, 5 nm in **g**, and 2 nm in **h**

Figure 2e and f present the optical microscope image and the corresponding AFM image of the as-grown AB stacking bilayer $MoS_2$ samples, respectively. The continuous height steps across from the substrate surface to the first monolayer and then to the second monolayer with each step height of ~0.7 nm reflect the nature of AB stacking bilayer $MoS_2$ films[6]. The AFM studies show that all bilayer domains including complete AA stacking, incomplete AA stacking with steps and AB stacking exhibit a smooth surface. The representative SEM images of AA and AB stacking bilayer $MoS_2$ domains are also presented in Supplementary Figure 7, further demonstrating the bilayer feature. The crystalline structure of the AA stacking $MoS_2$ bilayer films was further investigated by TEM, as shown in Fig. 2g (low resolution) and 2h (high resolution). Both images indicate the honeycomb lattice with hexagonal symmetry, and the FFT pattern shown in Fig. 2i clearly reveal the hexagonal symmetry of the [001] zone plane of $MoS_2$ lattice structures. The bilayer characteristic can be identified from the folding edge of the transferred thin film, as

shown in the inset of Fig. 2g. The lattice plane spacing ($d_{[100]}$) is calculated to be 0.27 nm, in good agreement with the reported values[47,48]. The fact that only AA and AB stacking $MoS_2$ bilayer grains were obtained in experiment means that they are the most stable ones, and this is in good agreement with the previously calculated results[37,49] and experimental results[37,38,50]. The detailed mechanism behind this can be found in Supplementary Information using extensive analysis of literature as well as theoretical calculations. Taking the reverse flow epitaxy method a step further, we were also able to grow $MoS_2$ trilayers including AAA, AAB, ABB as well as ABA stacking orders and even multilayers as shown in Supplementary Note II and Supplementary Figure 8 although the growth parameters require further optimization.

We further conducted micro-Raman and micro-PL studies to explore the optical properties of the as-grown bilayer $MoS_2$ grains, and the representative results on big $MoS_2$ bilayer flakes with a size of ~100 μm (including AA and AB stacking) are

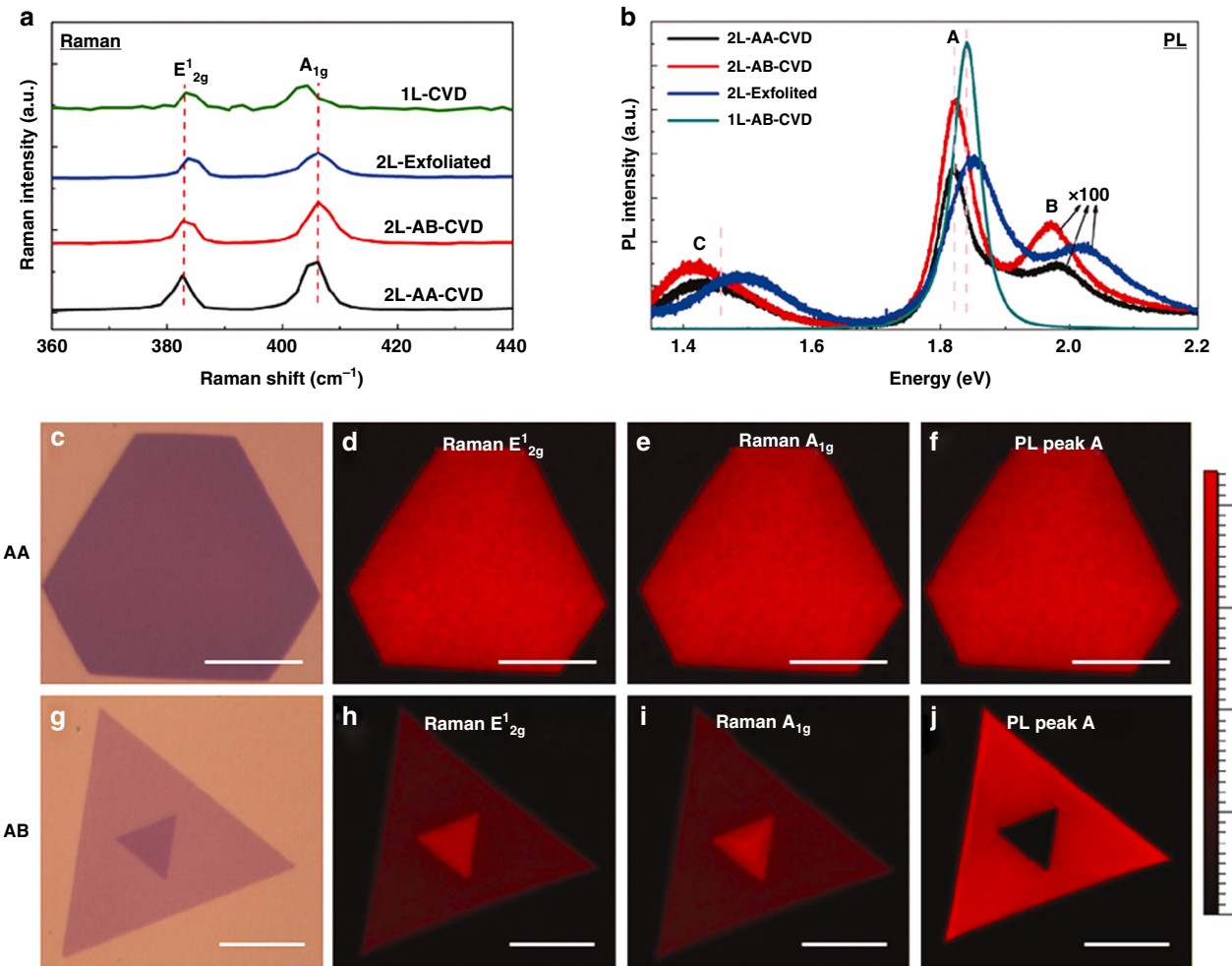

**Fig. 3** Raman and PL studies of the bilayer $MoS_2$ crystals. **a** Raman and **b** PL spectra of the as-grown AA and AB stacking bilayer $MoS_2$, CVD-grown monolayer $MoS_2$ and mechanically exfoliated bilayer $MoS_2$, where the PL signal of all bilayer samples were enlarged by 100 times in order to compare with that of monolayer one. Optical image (**c**) the Raman peak intensity mappings of $E^1_{2g}$ (**d**) and $A_{1g}$ (**e**) as well as the PL intensity mapping of peak A (**f**) of the as-grown complete AA stacking bilayer $MoS_2$. **g** Optical image, the Raman peak intensity mappings of $E^1_{2g}$ **h** and $A_{1g}$ **i** as well as the PL intensity mapping of peak A **j** of the as-grown AB stacking bilayer $MoS_2$. The scale bar is 50 μm

presented in Fig. 3. More results on small flakes are provided in Supplementary Note III. Figure 3a compares the Raman spectra of the as-grown AA and AB stacking bilayer $MoS_2$, CVD-grown monolayer $MoS_2$ and mechanically exfoliated bilayer $MoS_2$, respectively. The $A_{1g}$ peak of all the bilayer samples has a distinct blue shift in comparison with that of the monolayer sample due to the interlayer coupling[51,52]. The location of both $A_{1g}$ and $E^1_{2g}$ peaks are almost the same between the as-grown stacking bilayer and the mechanically exfoliated one, suggesting a similar situation of the Van der Waals force interactions between the upper layer and the bottom layer. The frequency difference for both the as-grown AA and AB stacking bilayer is estimated to be 22.2 cm$^{-1}$, quite similar to that (21.9 cm$^{-1}$) of the mechanically exfoliated one but larger than that (20.5 cm$^{-1}$) of the CVD-grown monolayer one[46,53]. The Raman peak intensity mappings of $E^1_{2g}$ and $A_{1g}$ of the as-grown complete AA stacking bilayer $MoS_2$ shown in Fig. 3d, e are very uniform, confirming the uniform in-plane stress and interaction between layers, while those of the as-grown AB stacking one shown in Fig. 3h, i reveal the clear boundary between the monolayer and bilayer regions, as observed from their optical images.

The PL spectra of all the bilayer samples presented in Fig. 3b exhibit three dominant peaks at ~1.8, ~2.0 and ~1.5 eV, labeled as

A, B and C, respectively, in great contrast to the prominent single and strong peak at around 1.8 eV of the monolayer sample. Both A and B peaks represent the direct gap transition[38]. All the bilayer samples show a greatly suppressed PL intensity (at least 100 times weaker) compared to the monolayer one because of the transition from a direct bandgap (monolayer) to an indirect one (bilayer)[5]. For the as-grown complete AA stacking sample, the PL intensity mapping of peak A presented in Fig. 3f is also quite uniform, matching well with the Raman and AFM results. For the as-grown AB stacking sample, however, the PL intensity mapping of peak A presented in Fig. 3j displays a distinct boundary between the monolayer and bilayer regions because of the different PL efficiency between direct bandgap (monolayer) and indirect bandgap (bilayer). Both Raman and PL mappings of the incomplete AA stacking bilayer $MoS_2$ with steps also have a strong contrast between the monolayer and the bilayer areas, as shown in Supplementary Figure 9. We also performed similar Raman and PL mapping studies on small $MoS_2$ bilayer flakes (including AA and AB stacking) and the results are shown in Supplementary Figures 10 and 11, reflecting the same properties of the small flakes as those of the big flakes.

The PL characteristics of the as-grown AA and AB stacking bilayer crystals can be further demonstrated by the density

functional theory (DFT) calculation. The van der Waals interaction is very important to describe the interlayer interaction of the bilayer $MoS_2$ crystals with AA or AB stacking. Many correction methods such as DFT-D2 correction[54] and DFT-TS correction[55] were developed to add the van der Waals interactions. Here, we adopt the Grimme's DFT-D2 dispersion correction approach to describe van der Waals interactions in all models and the reason can be found in Supplementary Note IV. The calculated energy band diagrams for $MoS_2$ monolayer, AA stacking $MoS_2$ bilayer, and AB stacking $MoS_2$ bilayer are shown in Fig. 4a–c, respectively. It is clear that $MoS_2$ monolayer has a direct bandgap, corresponding to the ultra-strong and single PL peak (A), while both AA stacking $MoS_2$ bilayer and AB stacking $MoS_2$ bilayer are of indirect bandgap with three electronic transitions from conduction band to valence band, corresponding to three weak PL peaks namely A, B, and C, respectively. The emergence of peak B can be attributed to the band splitting caused by spin–orbit coupling, and peak C is related to the indirect gap transition[35,40]. Figure 4d–i present side view and top view in ball-and-stick model as well as top view in MITSUBISHI column model of the atomic structures for AA and AB stacking $MoS_2$ bilayer crystals, respectively. The AA stacking (3R-like phase) lattice structure (Fig. 4d–f) consists of three staggered layers, where the Mo atoms of top layer reside upon the S atoms of the bottom layer, while both the Mo atoms of the bottom layer and the S atoms of the top layer reside upon the hollow[35,56]. The AB stacking (2H phase) lattice structure (Fig. 4g–i) is more simple, where the S (Mo) atoms of each layer reside upon the Mo (S) atoms of the other layer[38,50]. The theoretical reason behind the occurrence of both AA and AB stacking orders in the CVD-grown samples can be found in Supplementary Figure 12 and Supplementary Table 1.

**Electrical performance of the bilayer MoS$_2$-based FETs**. The back-gated FET devices were also fabricated on the as-grown AA and AB stacking bilayer $MoS_2$ samples to measure their electronic characteristics, and four devices have been tested for each group in order to explore consistency in electrical performances. Figure 5a-b and 5c-d present the representative transfer characteristics and output characteristics for AA and AB stacking bilayer $MoS_2$ FET devices, respectively. The electrical performances of the other three devices can be found in Supplementary Figures 13 and 14 (Supplementary Note V), respectively, for AA and AB stacking samples. For both groups, the transfer characteristics show typical n-type behavior since the source-drain current increase with the increase in $V_g$ from −60 to 60 V, which is the same as that commonly observed in monolayer one[46,57]. It is interesting that the output characteristics ($I_{ds} − V_{ds}$) of AA stacking devices (Fig. 5b and Supplementary Figure 13) are more likely to show linear and symmetrical relationships, which are similar to the performances of CVD-grown monolayer devices (see Supplementary Figure 15), while those of AB stacking devices (Fig. 5d and Supplementary Figure 14) show nonlinear and asymmetrical behaviors, indicating obvious Schottky contact and rectification characteristic. The differences between these two kinds of devices may lie in their different stacking orders and their different contact situations with metal electrodes. Both the electrode configuration schematics in Supplementary Figure 16a, b as well as the output characteristics ($I_{ds} − V_{ds}$) measured by switching source and drain electrodes (Supplementary Figure 16c) prove that both electrode contacts for AB stacking devices are of Schottky type, which constitute two back-to-back junctions and is thus responsible for the rectifying phenomena[58].

We further calculate the mobility and on/off ratio values for both groups of devices. The field effect mobility is calculated by the following equation:

$$\mu_{FET} = \left( \frac{L}{W C_g} \right) \left( \frac{dI_{DS}}{dV_g V_{DS}} \right) \tag{1}$$

where $\frac{dI_{DS}}{dV_g V_{DS}}$ is the slope, $L$ and $W$ are the channel length and

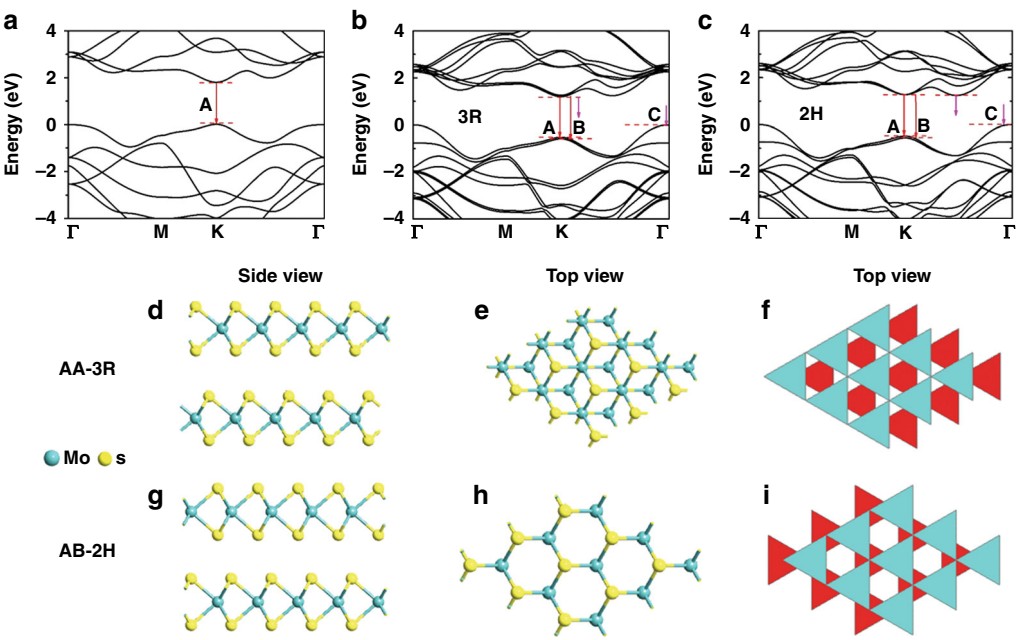

**Fig. 4** Atomic structure and band structure of the bilayer $MoS_2$ crystals. The calculated energy band structures of **a** $MoS_2$ monolayer, **b** AA-stacking $MoS_2$ bilayer and **c** AB stacking $MoS_2$ bilayer. **d** The side view and **e** top view in ball-and-stick model as well as **f** top view in MITSUBISHI column model of the atomic structures for AA stacking $MoS_2$ bilayer; the counterpart views **g**, **h**, and **i** for AB stacking $MoS_2$ bilayer. In **d**, **e**, **g**, and **h**, the blue solid spheres represent Mo atoms and the yellow ones S atoms. In **f** and **i**, the red and blue triangles stand for the bottom and upper $MoS_2$ layer, respectively, where Mo atoms locate at the center of each triangle and S atoms at the apex of each triangle

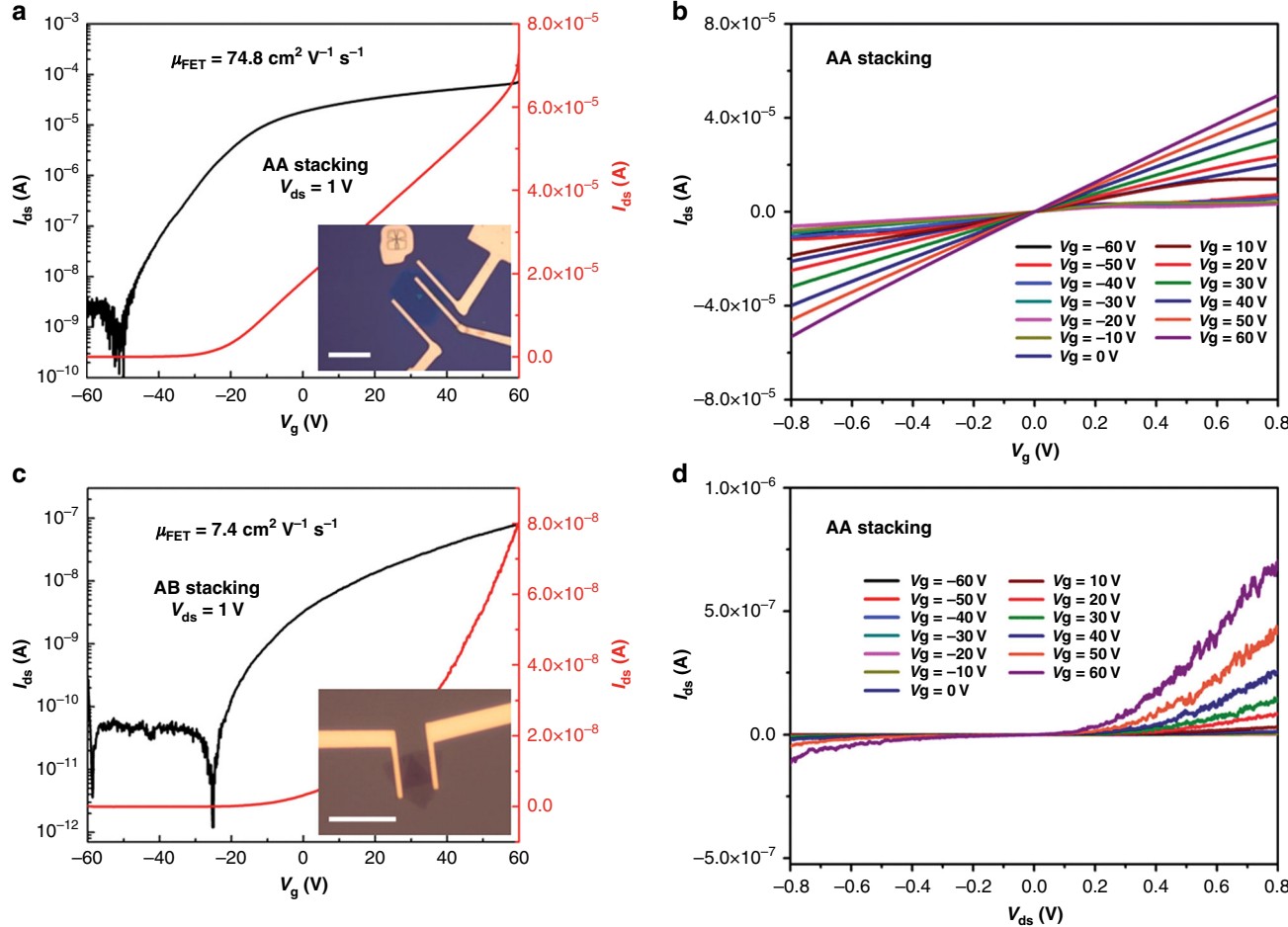

**Fig. 5** Electrical properties of the bilayer $MoS_2$ FET devices. The representative transfer characteristics ($I_{DS} - V_g$) and output characteristics ($I_{DS} - V_{DS}$) for **a**, **b** AA and **c**, **d** AB stacking bilayer $MoS_2$ FET devices. Insets in **a** and **c** are the optical images of the AA and AB stacking bilayer $MoS_2$ devices, respectively. Scale bars are 10 μm in the insets of **a** and **c**

width, respectively, and $C_g$ is the capacitance per unit area between the channel and the back gate[59]. The AA stacking samples possess an on/off ratio in the order of ~$10^4$ and exhibit high mobility values in the range of 38–75 $cm^2 V^{-1} s^{-1}$, which outperform those reported values of monolayer $MoS_2$ FETs[53,60]. In contrast, the AB stacking ones show rather lower mobility values in the range of 1–7 $cm^2 V^{-1} s^{-1}$. Such great differences in mobility may be related to the different stacking order, as well as the different contact characteristics. The sizeable Schottky barriers for AB stacking-based FETs may limit the current output and thus lead to lower, extrinsic mobility values since the electrical response was measured in a two contact configuration as previously reported[61]. The carrier mobility of the FETs can usually be further improved by the deposition of a high-$\kappa$ dielectric layer (i.e. $HfO_2$) in a top gate configuration[57] or using the four terminal-configurations to evaluate the intrinsic carrier mobility as reported[61]. The different energy band structures between AA and AB stacking induced by the different stacking order may also result in some discrepancy in mobility values.

**Universality in other TMD bilayer crystals**. Our CVD approach can also be expanded to synthesise diverse 2D TMD bilayer crystals like $WS_2$, ternary alloy $Mo_{1-x}W_xS_2$, and quaternary alloy $Mo_{1-x}W_xS_{2(1-y)}Se_{2y}$. By simply replacing $MoO_3$ sources by $WO_3$ sources and slightly adjusting the working temperature,

we can obtain large area $WS_2$ bilayer crystals using the same sequential two-stage CVD strategy. The representative optical images, AFM surface morphologies, Raman and PL spectra, as well as Raman and PL intensity mappings presented in Supplementary Note VI and Supplementary Figure 17 reflect the high uniformity and high quality of the as-grown $WS_2$ bilayer crystals. By using both $MoO_3$ and $WO_3$ as the transition metal sources and both S and Se powder as the sulfide sources and at the same time placing them in separate positions (with different working temperatures) to enable them simultaneously evaporate, we can also achieve large area quaternary alloy $Mo_{1-x}W_xS_{2(1-y)}Se_{2y}$ bilayer crystals. One can also observe the high uniformity and high quality of the as-grown quaternary bilayer crystals from the representative optical images and AFM surface morphologies shown in Fig. 6a–e. The Raman spectra shown in Fig. 6f display four prominent peaks at 271, 359, 376, and 404 $cm^{-1}$. Among them, the two peaks at 376 and 404 $cm^{-1}$ are close to $E^1_{2g}$ and $A_{1g}$ of pure $MoS_2$ bilayer crystals, respectively, the peak at 359 $cm^{-1}$ is close to $E^1_{2g}$ of $WS_2$ but with a slight blue shift, and the peak at 271 $cm^{-1}$ locates between $A_{1g}$ of $WSe_2$ (263 $cm^{-1}$) and $E^1_{2g}$ of $MoSe_2$ (282 $cm^{-1}$). Similar to the cases of monolayer and bilayer $MoS_2$ crystals, the $Mo_{1-x}W_xS_{2(1-y)}Se_{2y}$ bilayer crystals show a greatly suppressed PL intensity with three peaks corresponding to direct and indirect gap transitions in comparison with the prominent single and strong peak of the monolayer ones, as presented in Fig. 6g. The XPS data displayed in Fig. 6h supports the coexistence of Mo, W, S, and Se elements, indicating the quaternary characteristic of such

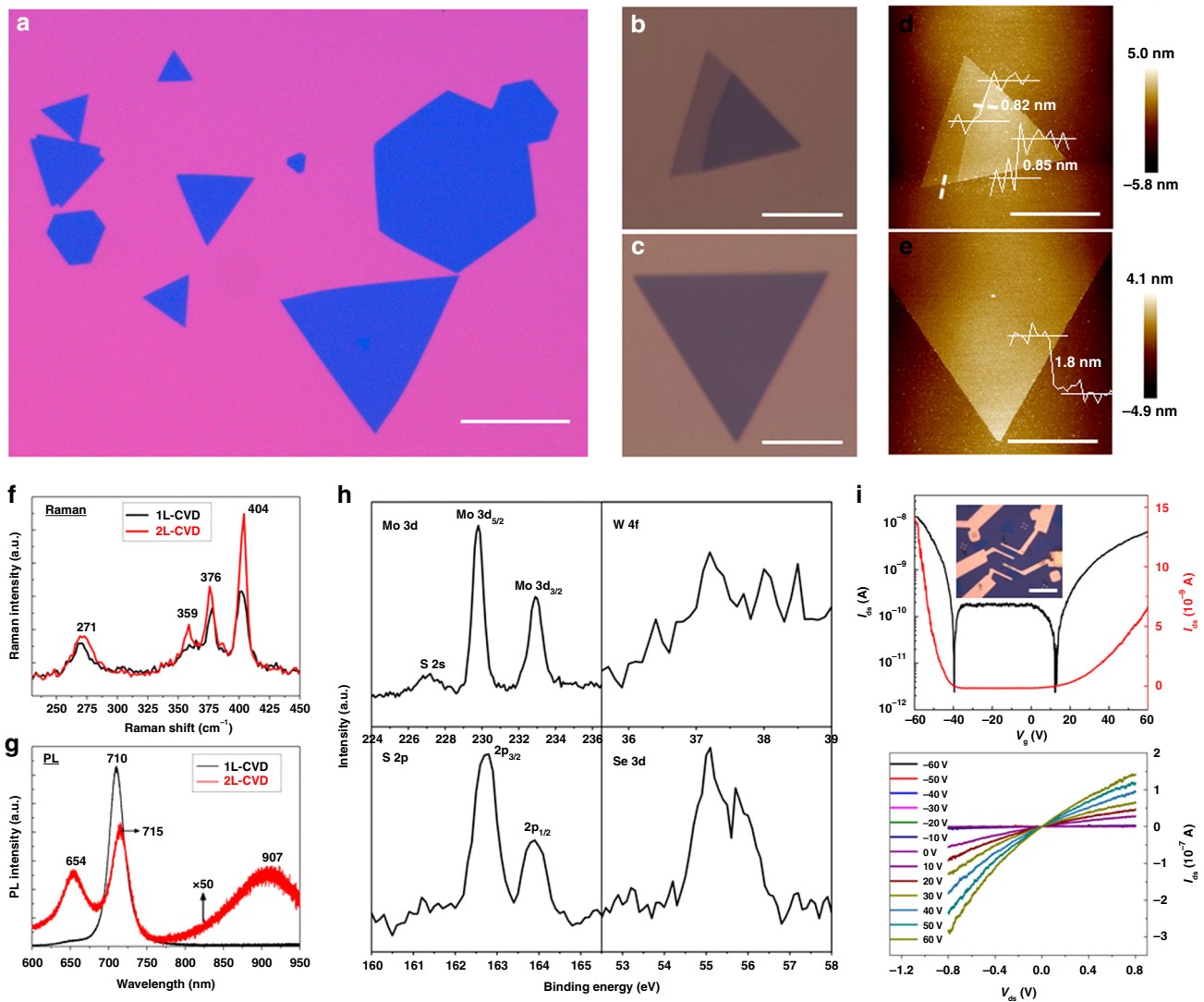

**Fig. 6** Characterization of the quaternary alloy $Mo_{1-x}W_xS_{2(1-y)}Se_{2y}$ bilayer crystals. **a–c** The representative optical images as well as **d,e** AFM surface morphologies of the as-grown quaternary alloy $Mo_{1-x}W_xS_{2(1-y)}Se_{2y}$ bilayer crystals. **f** Raman spectra and **g** PL spectra of the as-grown quaternary alloy $Mo_{1-x}W_xS_{2(1-y)}Se_{2y}$ bilayer and monolayer crystals. **h** XPS spectra of the as-grown quaternary alloy $Mo_{1-x}W_xS_{2(1-y)}Se_{2y}$ bilayer crystals. **i** Transfer and output characteristics of the back-gated FET devices fabricated on the $Mo_{1-x}W_xS_{2(1-y)}Se_{2y}$ bilayer samples. The inset is one typical optical image of the FET device. Scale bars, 100 µm in **a**, 20 µm in **b–e**, 10 µm in the inset of **i**

bilayer crystals. Furthermore, the back-gated FET devices fabricated on the $Mo_{1-x}W_xS_{2(1-y)}Se_{2y}$ bilayer samples exhibit a bipolar characteristic (Fig. 6i), which may be attributed to the fact that the quaternary bilayer crystals are partly composed of n-type $MoS_2$ and p-type $WSe_2$. Furthermore, the composition of ternary alloy $Mo_{1-x}W_xS_2$ and quaternary alloy $Mo_{1-x}W_xS_{2(1-y)}Se_{2y}$ can be tuned by adjusting the ratio of these source powders, as demonstrated by the preliminary result in Supplementary Note VII and Supplementary Figure 18.

## Discussion
Together, we have developed a reverse-flow chemical vapor epitaxy strategy to synthesize high-quality TMD bilayer crystals with high yield, large size and high controllability. Customized temperature profiles and reverse gas flow help activate the first layer without introducing new nucleation centers leading to near-defect-free epitaxial growth of the second layer from the existing nucleation centers. A series of TMD bilayer crystals including pure $MoS_2$ and $WS_2$, ternary $Mo_{1-x}W_xS_2$ and quaternary $Mo_{1-x}W_xS_{2(1-y)}Se_{2y}$ were prepared with variable structure

configurations and tunable electronic/optical properties. The as-grown bilayer crystals confirmed by AFM and TEM studies all possess high uniformity in terms of Raman and PL spectra mappings. $MoS_2$ bilayer crystals with AA and AB stacking structures can be obtained by adjusting the growth temperature of the second monolayer. AA stacking $MoS_2$ bilayer crystals exhibit higher field-effect mobility values than monolayer ones. The FET devices based on quaternary alloy $Mo_{1-x}W_xS_{2(1-y)}Se_{2y}$ bilayer crystals display a bipolar characteristic because of the intrinsic coexistence of n-type $MoS_2$ and p-type $WSe_2$. Our studies thus provide a robust, potentially universal approach for the synthesis of large-size TMD bilayer single crystals, representing a highly promising materials system for device demonstrations at the limit of single atomic thickness and for fundamental studies arising from the interlayer van der Waals interactions, such as screening effect, quantum confinement, structural symmetry and so on.

## Methods
**CVD growth of TMD bilayer**. TMD bilayer crystals were grown by the modified sequential two-stage CVD system consisting of heating zone 1 (upstream) and

heating zone 2 (downstream). The reverse-flow chemical vapor epitaxy process was intentionally divided into two growth stages: A-B stage stands for the growth of first monolayer, C-D stage (relatively higher working temperature) represents the vertically epitaxial growth of second monolayer, while B-C stage corresponds to the growth swing stage between the first and second layers. The system is modified so that the gas flow direction can be switched (reversed) during the growth swing stage (B-C stage). A forward gas flow of 50 sccm $N_2$ was introduced and then maintained from the source to the substrate during the whole growth process except B-C stage, where a reverse gas flow consisting of 50 sccm $N_2$ and 5 sccm $H_2$ from the substrate to the source was introduced.

**$MoS_2$ bilayer**. Fifteen milligrams of $MoO_3$ powder (99.9%) was put in a ceramic boat in the center of heating zone 2 and excessive S powder (100 mg, 99.9%) were put in the heating zone 1 (upstream) with a distance of about 38 cm away from the $MoO_3$ powder. The Si substrates with 300 nm $SiO_2$ ($SiO_2$/Si) were placed facing down above the $MoO_3$ powder. The heating zone 2 ($MoO_3$ precursor) was heated but the heating zone 1 (S precursor) was turned off intentionally so that the latter would become hotter as the former was heated up because of thermal radiation. The distance of 38 cm between $MoO_3$ and S powders ensure the simultaneous sublimation of both at growing stages. The heating zone 2 was initially heated to 700 °C in 100 min and then maintained at 700 °C for 10 min for the growth of first $MoS_2$ monolayer (A-B stage). Afterwards, the temperature of heating zone 2 was further increased to 750 or 800 °C in 5 min (B-C stage) and then kept for 20 min for the vertically epitaxial growth of second $MoS_2$ monolayer (C-D stage). Experimental results showed that 750 °C is suitable for AA stacking $MoS_2$ bilayer crystals while 800 °C is optimal for AB stacking ones.

**$WS_2$ bilayer**. Similar to the case of $MoS_2$ bilayer, 25 mg $WO_3$ powder and 100 mg S powder were used as the sources and their distance was fixed at 42 cm to ensure the simultaneous sublimation at growing temperatures. The heating zone 2 was initially heated to 800 °C in 100 min and then maintained at 850 °C for 10 min for the growth of first $WS_2$ monolayer (A-B stage). Afterwards, the temperature of heating zone 2 was further increased to 900 °C in 5 min (B-C stage) and then kept for 20 min for the vertically epitaxial growth of second $WS_2$ monolayer (C-D stage). The other experimental parameters were the same as those in the growth case of $MoS_2$ bilayer.

**$Mo_{1-x}W_xS_{2(1-y)}Se_{2y}$ bilayer**. Ten milligrams of $WO_3$ powder was placed at the center of heating zone 2 while 15 mg $MoO_3$ powder at the downstream of heating zone 2 with a suitable distance of 8–10 cm away from the $WO_3$ source. Hundred milligrams of S powder and 25 mg Se powder were put in the heating zone 1 with distances of 42 and 38 cm away from the $WO_3$ powder, respectively. The selected distances between the sources enable the simultaneous sublimation of them at growing stages. The heating zone 2 was initially heated to 900 °C in 100 min and then maintained at 900 °C for 10 min for the growth of first $Mo_{1-x}W_xS_{2(1-y)}Se_{2y}$ monolayer (A-B stage). Afterwards, the temperature of heating zone 2 was further increased to 950 °C in 5 min (B-C stage) and then kept for 20 min for the vertically epitaxial growth of second $Mo_{1-x}W_xS_{2(1-y)}Se_{2y}$ monolayer (C-D stage). The other experimental parameters were the same as those in the growth case of $MoS_2$ bilayer.

**Microscopy and microanalysis**. The Raman and PL spectra characterizations were measured using a Renishaw LabRAM Invia micro-Raman system with 532 nm excitation laser at room temperature and in atmospheric environment. The testing laser spot size was about 1 μm by using a ×50 objective and the laser power was kept below 0.3 mW. The thickness of TMD bilayer and monolayer crystals was measured by AFM (Bruker model: Dimension ICON). The elemental composition of the as-grown TMD bilayer crystals was identified by XPS (Thermor Scientific Escalab 250Xi) in which the Al-ka (1486.6 eV) was used as source. TEM characterizations were performed with the Jeol JEM-ARM200F equipped with a cold field emission electron source and a probe-Cs-corrector at 200 kV accelerating voltage. The electronic characteristics of the TMD bilayer FETs were tested by a Keithley 2612A Source Meter in vacuum (~1 Pa) at room temperature.

**First principles calculation**. The geometry optimizations and the energy band structure calculations of the three kinds of $MoS_2$ models, the monolayer one, the bilayer ones with AA and AB stacking orders, were calculated through First-principles based on DFT, which were implemented in the Atomistic-ToolKit (ATK) version 2017.0. The Perdew–Burke–Ernzerhof (PBE) of generalized gradient approximation (GGA) was used for studying the band structures. Besides, pseudopotentials of $MoS_2$ using the Hartwingster–Goedecker–Hutter scheme with Tier 4 basis set. The density mesh cutoff was set as 75 Hartree to achieve the energy convergence of 10−5 eV and the force convergence of 0.01 eV/Å on each atom. The Brillouin zone was sampled by $9 \times 9 \times 1$ and $25 \times 25 \times 1$ Monkhorst–Pack $K$-point mesh for structure optimization and electronic band calculation. A vacuum spacing of 15 Å was used to avoid interaction between the calculated individual structures. In order to prove the validity of this method, we also benchmark our calculated results of monolayer $MoS_2$ with other available publications which can be found in Supplementary Information.

**FET device fabrication**. The field effect transistor (FET) devices were fabricated with the as-grown TMD bilayer samples just on $SiO_2$/Si substrates, 5 nm-thick Ni and 50 nm-thick Au (Ni/Au) patterned by standard electron beam lithography (EBL) were used as the source and drain electrodes. The $SiO_2$ layer with a thickness of about 300 nm acted as the dielectric layer and the P-doped silicon substrate served as the back gate electrode.

## Data availability

The data that support the findings of this study are available from the corresponding author on reasonable request.

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

## Acknowledgements

This work is partially supported by the National Nature Science Foundation under Grants 11704159, 61404061 and 61422503, the Natural Science Foundation of Jiangsu Province, China under Grants BK20170167 and BK20140168, the Fundamental Research Funds for the Central Universities of China under Grants JUSRP51726B, the 111 Project under Grant B12018, as well as by the Australian Research Council (ARC) and CSIRO's Science Leaders Program.

## Author contributions

S.X. and X.Z. conceived the idea, designed, and performed the experiments, and jointly with H.N., X.G., and K.O. analyzed the data and interpreted the results. X.Z. and X.W. conducted the TEM characterizations. H.N. and Z.N. conducted the electrical measurements. X.Z. and A.D. conducted the first principles calculation. S.X., X.Z., H.N., K.O., and X.G. co-wrote the paper. All authors discussed the results and commented on the manuscript.

## Additional information

**Competing interests:** The authors declare no competing interests.

