## [Peer Review File · Nature Communications]

Reviewers' comments:

Reviewer #1 (Remarks to the Author):

The major claims of the paper is to realize epitaxial growth of the second monolayer from the first monolayer by reverse-flow chemical vapor epitaxy. This study is important in terms of high-quality, large-size TMD bilayer crystals, which is still a challenge. The utilized method is described as reverse-flow chemical vapor epitaxy, which is suggested to provide near-defect-free epitaxial growth of the second layer. Control of nucleation centers is of significance to realize large size and high quality TMD crystals.

Monolayer TMD growth is of importance because of the direct band gap providing the materials to be used in optoelectronic applications. However, bilayer TMDCs are also critical because of their potential use in transistors and logic circuits with relatively higher mobilities compared to the monolayer based devices.

The novelty of this study lies in the usage of a reverse hydrogen flow for bilayer growth so that the excessive nucleation sites are reduced. However, this technique was already introduced before (demonstrating the growth of stacked heterostructures). Although the relevant reference is given, sufficient explanation is not provided. "Robust epitaxial growth of two-dimensional heterostructures, multiheterostructures, and superlattices" Z. Zhang et al., *Science* 10.1126/science.aan6814 (2017)

On the other hand, still using the technique for bilayer growth can be considered as novel, which will be of interest to the researcher in the community.

Fig 1 a needs modification because it does not correctly show the modified sequential two-stage thermal CVD process: but instead it shows a conventional CVD set up.

The conclusions are sufficiently original; explanation of atomic structure and band structure of the AA and AB stacking bilayer MoS₂ provides clarity on the formation of these bilayer structures. FET fabrication and the related results show that mobility values are considerably improved with respect to the ones based on monolayers. However, the benefit of bilayer TMD growth can mostly be explained by the improved mobility of the transistors and in the manuscript, the issues to improve the mobility to higher values is not sufficiently discussed or addressed (e.g. using a four-terminal configuration, *Appl. Phys. Lett.* 102, 123105 (2013); <https://doi.org/10.1063/1.4799172>)

As a result, although there are some minor issues, this research work is presented in a convincing way to influence thinking in the field.

Reviewer #2 (Remarks to the Author):

In this manuscript author has grown large size 2D materials bilayers by reverse flow chemical epitaxy method. AA and AB bilayer flakes were grown by tuning the growth temperature of second layer. The reviewer has following queries

1. AFM gives information about morphology but not about staking. Author should provide the clear evidence of AA and AB staking.
2. Second layer in AA staking not grown on the edges of the flake (1st layer), why?
3. Why is the TMD layer not grown on SiO₂ during the second layer growth?
4. If the reverse flow is deciding the stake type then grow third monolayer and show the AAA and ABA stakes growth.
5. There is possibility in AB stake (high temp. growth) the stress is different from AA stake (low temp. growth) and which could be a deciding factor for materials property. What is your comment on stress?
6. Author has grown big flakes but AFM, Raman and PL measurements were done on small flakes,

why? are the properties of small and big flakes same? Provide some evidence.

7. In figure 3(e), PL brightness (intensity) should be higher at the edge of flakes due to single layer but not observed, why? However, it can be clearly seen in fig 3(h) of AB stake.

8. The I_{ds} - V_{ds} plot is linear for AA and rectifying for AB stakes, why?

9. The conduction is through top layers so I-V behaviour should be same. Author should provide a clear conduction mechanism.

10. Compare the I_{ds} - V_{ds} behaviour of mono and bi layers.

11. Why mobility is low in AB stakes? The mobility should be same.

12. Could you control the value of x and y in $Mo_{1-x}W_xS_2(1-y)Se_y$?

Reviewer #3 (Remarks to the Author):

This is a very interesting work about the growth of transition metal dichalcogenides bilayer single crystals using a method of reverse flow chemical vapor epitaxy. A series of TMD bilayer crystals including MoS₂ and WS₂, ternary $Mo_{(1-x)}W_xS_2$ and quaternary $Mo_{(1-x)}W_xS_2(1-y)Se_y$ are synthesized with variable structural configurations and tunable electronic and optical properties. In addition, TMDC bilayer crystals with AA and AB stacking structures can be obtained by adjusting the growth temperature of the second monolayer. These results representing a promising TMDC materials system for fundamental studies and technological applications. The article is suggested to be publish after revised.

Question 1: Can you show more information about the growth under different temperatures (e.g., 720°C, 760°C, etc.)? In this way, the relationship between temperature and the epitaxial growth behaviour of the second monolayer can be demonstrated more clearly.

Question 2: It's necessary to show relationship between the size of the second layer MoS₂ and the growth time at 750°C? This will gives a more clearer picture of the growth process.

Question 3: As we can see in the Fig 2(e), the second layer of A-B stacking MoS₂ cannot completely cover the first layer . What's the reason? Can you get a full coverage A-B stacking MoS₂?

Question 4: In Fig. 2(i), In addition to the normal white diffraction spots, we can also find some unnormal small points that are not obvious. What's the reason?

Question 5: Can you get trilayer or more layer 2D-TMD through this method?

Reviewer #4 (Remarks to the Author):

The authors present a detailed study and description of a novel experimental procedure based on reverse-flow chemical vapor epitaxy which allows them to produce large-size TMD bilayer crystals of high quality. The quality of the samples produced via this method is then tested and the samples are characterized by a variety of methods and techniques, indicating the successful synthesis of high quality crystals. In addition, the authors have performed electrical transport studies which show that the bilayer MoS₂ samples have better electronic performance than the monolayer counterparts. To support their experimental observations, the authors have performed DFT calculations, from which they have generated band structure plots for bilayer MoS₂ samples with AA and AB stacking. Although their theoretical model is simplified, it seems to be sufficient to support their experimental PL findings. Since they are studying bilayer structures, however, it is important to include van der Waals interactions, which have been ignored in this model system. In addition, the authors should present additional details on the theoretical methods which are

needed to reproduce the model. For example, they need to report the lattice constant for MoS₂, and details about force convergence. There is also a typo in the description of the PBE functional, it should read "Perdew-Burke-Ernzerhof".

Overall the manuscript has high value as far as the experimental method that is presented here is concerned. The authors claim that the method can be universally applied to produce bilayers of different TMD compounds, which is valuable for the design of advanced electronics. The theory part is simple, and the bilayer model of TMDs has been extensively studied from other groups. Therefore, it doesn't add much value to the article. The authors could have used band structure plots that are available in the literature to support their experimental observations.

Minor comment: Figure 1 has (a), (b) and (c) parts, but the authors refer to Fig. 1d on page 5, line 123 and page 6, line 138.

Response to the comments and suggestions of Reviewer #1

General comment: *The major claims of the paper is to realize epitaxial growth of the second monolayer from the first monolayer by reverse-flow chemical vapor epitaxy. This study is important in terms of high-quality, large-size TMD bilayer crystals, which is still a challenge. The utilized method is described as reverse-flow chemical vapor epitaxy, which is suggested to provide near-defect-free epitaxial growth of the second layer. Control of nucleation centers is of significance to realize large size and high quality TMD crystals. Monolayer TMD growth is of importance because of the direct band gap providing the materials to be used in optoelectronic applications. However, bilayer TMDCs are also critical because of their potential use in transistors and logic circuits with relatively higher mobilities compared to the monolayer based devices. The conclusions are sufficiently original; explanation of atomic structure and band structure of the AA and AB stacking bilayer MoS₂ provides clarity on the formation of these bilayer structures. FET fabrication and the related results show that mobility values are considerably improved with respect to the ones based on monolayers.*

Response/corrections: We are grateful for the positive comments given by the Reviewer. We have made a substantial improvement in the revised manuscript according to the Reviewers' comments and hope that the manuscript is now suitable for publication.

Comment 1. *The novelty of this study lies in the usage of a reverse hydrogen flow for bilayer growth so that the excessive nucleation sites are reduced. However, this technique was already introduced before (demonstrating the growth of stacked heterostructures). Although the relevant reference is given, sufficient explanation is not provided. "Robust epitaxial growth of two-dimensional heterostructures, multiheterostructures, and superlattices" Z. Zhang et al., Science 10.1126/science.aan6814 (2017). On the other hand, still using the technique for bilayer growth can be considered as novel, which will be of interest to the researcher in the community. Fig 1 a needs modification because it does not correctly show the modified sequential two-stage thermal CVD process: but instead it shows a conventional CVD set up.*

Response/corrections: We thank the Reviewer for these helpful comments. The revised Fig. 1a and more detailed explanation are provided to explain the effect of the reverse hydrogen flow on the epitaxial growth of the second monolayer from the first monolayer in the revised manuscript. Please refer to the paragraph marked by red color on Page 5 of the

revised manuscript. We also copy this paragraph below for the Reviewer's convenience (references are numbered as in the revised manuscript).

The strikingly positive effect of such reverse N_2/H_2 flow during the temperature swing stage can be explained as follows. First, the reverse carrier gas flow can prevent unintended supply of chemical vapor source to eliminate the generation of new nucleation centers on the growth substrate and the as-grown first layer during the temperature rising process⁴⁰. Zhang et al.⁸ employed the same idea to prevent uncontrolled homogeneous nucleation and thus enabled highly robust epitaxial growth of diverse heterostructures, multi-heterostructures and superlattices from 2D atomic crystals. Second, the hydrogen flow could saturate the dangling bonds on the edge and at the surface of the as-grown first MoS_2 monolayer crystals, thus blocking the laterally epitaxial growth as reported by Jia et al⁴¹. The surface energy of the edge-terminated structure is considerably higher compared to that of the as-grown flat basal-plane structure^{42,43}. Consequently, the second monolayer is more likely to deposit on the as-grown monolayer surface. At the same time, H_2 can slightly etch away emerging nucleation points on the growth substrate and thereby reduce the wettability of the growth substrate⁴⁴. As such, the source vapor during the C-D stage can be easier to transfer through the substrate surface and have more possibility to reach the surface of the as-grown first monolayer. With a suitable carrier gas flow rate, the source vapor can have enough kinetic energy to reach the surface center of the as-grown first monolayer where there is an initial nucleation to begin the growth of the second layer⁴⁵. Therefore, the second monolayer crystals prefer to grow epitaxially and homogeneously on the activated nucleation centers of the first monolayer, finally promoting the growth of MoS_2 bilayer crystals, as shown in Fig. 1b.

As mentioned above, we have also modified Fig. 1a to correctly show the modified sequential two-stage thermal CVD process and added the corresponding description in the revised manuscript. Please refer to the revised Fig. 1a and the sentences marked by red color on Page 4 of the revised manuscript. We also copy the sentences together with Fig. 1 below for the Reviewer's convenience.

Both sides of the CVD tube are equipped with gas inlet and outlet. The direction of gas flow can be switched by simultaneously turning on gas valves 1 and 4 (or gas valves 2 and 3) and turning off gas valves 2 and 3 (or gas valves 1 and 4).

Fig.1 (a) Experimental setup and the temperature program of the modified sequential two-stage thermal CVD process: A-B stage stands for the growth of first layer, C-D stage represents the growth of second layer, while B-C stage corresponds to the growth swing stage for the first and second layer. A reverse N_2/H_2 flow from the substrate to the source was introduced during the temperature swing stage (B-C stage); (b) a representative optical image of the as-grown bilayer MoS_2 crystal grains; (c) schematic diagram of the reverse-flow chemical vapor epitaxy process for bilayer MoS_2 . Different growing temperature at C-D stage can result in bilayer MoS_2 crystals with different stacking structures: 750 °C for AA stacking bilayer crystals and 800 °C for AB stacking ones.

Comment 2. However, the benefit of bilayer TMD growth can mostly be explained by the improved mobility of the transistors and in the manuscript, the issues to improve the mobility to higher values is not sufficiently discussed or addressed (e.g. using a four-terminal configuration, *Appl. Phys. Lett.* 102, 123105 (2013); <https://doi.org/10.1063/1.4799172>) Intrinsic carrier mobility of multi-layered MoS_2 field-effect transistors on SiO_2 . As a result, although there are some minor issues, this research work is presented in a convincing way to influence thinking in the field.

Response/corrections: We greatly appreciate this suggestion. We have discussed the issues to improve the mobility to higher values referring to these reports [*Appl. Phys. Lett.* 2013, 102, 123105; *Nat. Nanotechnol.* 2011, 6, 147]. Please refer to the paragraph marked by red color on Page 14 in the revised manuscript. References are numbered as in the revised manuscript.

The sizeable Schottky barriers for AB stacking based field-effect transistors may limit the current output and thus lead to lower, extrinsic mobility values since the electrical response was measured in a two contact configuration as previously

reported⁶¹. The carrier mobility of the FETs can usually be further improved by the deposition of a high- κ dielectric layer (i.e. HfO₂) in a top gate configuration⁵⁷ or using the four terminal-configurations to evaluate the intrinsic carrier mobility as reported⁶¹. The different energy band structures between AA and AB stacking induced by the different stacking order may also result in some discrepancy in mobility values.

Response to the comments and suggestions of Reviewer #2

Comment 1. AFM gives information about morphology but not about staking. Author should provide the clear evidence of AA and AB staking.

Response/corrections: We agree with the Reviewer that AFM indeed does not clearly determine the stacking and greatly appreciate this suggestion. The clear evidence of AA and AB stacking is provided in the revised Figs. 2c, 2e, 2d and 2f. It has been already proven that the orientation of each triangle is directly correlated with the microscopic crystal orientation of the MoS₂ layer, and one can determine the twist angle of a MoS₂ bilayer from the relative rotation of the two vertically stacked triangles [NPG Asia Mater. 2018, 10, e468; Nano Lett. 2015, 15, 8155; Nat. Commun. 2014, 5, 4966]. The bilayer structures with the twist angle $\theta=0^{\circ}$ are usually called AA stacking samples and those with the twist angle $\theta=60^{\circ}$ are called AB stacking samples. We have shown the twist angles in both the optical (Figs. 2c and 2e) and AFM (Figs. 2d and 2f) images for AA and AB stacking samples in the revised manuscript. In addition, such AA and AB stacking structures (with similar optical images and AFM images) have already been identified by the TEM [Adv. Mater. 2017, 29, 1604540; Nano Lett. 2015, 15, 8155; ACS Nano 2015, 12, 12246; Phys. Rev. B 2016, 93, 041420; Phys. Rev. B 2015, 91, 235202; Phys. Rev. Lett. 2013, 111, 106801] and the SHG test [Nat. Commun. 2014, 5, 4966]. Such AA and AB stacking structures can be corresponding to 3R-like and 2H phase crystals, respectively. In this work, we performed AFM measurements on AA stacking samples with no steps between the two layers, AA stacking samples with steps between the two layers and AB stacking samples, as shown in Figs. 2b, 2d and 2f. We have also revised the caption of Figure 2 (marked in red in the revised manuscript), which is copied below for the Reviewer's convenience. References are numbered as in the revised manuscript.

Fig.2 A representative optical microscope image (a) and a representative AFM surface morphology image (b) of the as-grown AA stacking bilayer MoS₂ samples with the maximum lateral size of even up to 300 μm ; a representative optical image (c) and a representative AFM surface morphology picture (d) of the AA stacking bilayer MoS₂ grains with distinct steps, where the twist angle from the relative rotation of the two vertically stacked triangles $\theta=0^{\circ}$ indicate the AA stacking order^{37,50,51}; a typical optical image (e) and a typical AFM surface morphology image (f) of the as-grown AB stacking bilayer MoS₂ samples, where the twist angle from the relative rotation of the two vertically stacked triangles $\theta=60^{\circ}$ indicate the AB stacking order^{37,50,51}; planar TEM images of the AA stacking bilayer MoS₂ crystals: low resolution (g) and high resolution (h) as well as the selected area electron diffraction (SAED) patterns (i); the inset in (g) shows the folded edge of MoS₂ bilayer films.

Comment 2. *Second layer in AA staking not grown on the edges of the flake (1st layer), why?*

Response/corrections: We thank the Reviewer for this comment. In our experimental results, there are two kinds of AA stacking samples: one is the complete AA stacking without any steps as shown in Figs. 2a and 2b, and the other is the incomplete AA stacking with distinct steps (in which the second layer is not grown on the edges of the first layer) as shown in Figs. 2c and 2d. This different outcome can be attributed to the different growth time of C-D stage (for the second layer growth). The representative optical images of AA stacking MoS₂ bilayer samples obtained under different growth time of C-D stage (for the second layer growth) are presented in Figure S5. When the growth time of C-D stage was shortened to less than 10 minutes (like 5 minutes), the epitaxial growth of the second monolayer is insufficient to fully cover the surface of the first monolayer so that the incomplete AA stacking MoS₂ bilayer grains with distinct steps may dominate on the substrate surface. However, when the growth time was kept for about 10 minutes, both complete and incomplete AA stacking samples can coexist on the substrate surface. When the growth time of C-D stage was kept for more than 12 minutes (like 15 and 20 minutes), the complete AA stacking bilayer samples without any steps can be predominately synthesized on the whole substrate surface. We have also added the corresponding discussion (marked by red colour) on Page 7 of the revised manuscript as well as on Pages 6 and 7 of SI. Figure S5 is also attached below for the Reviewer's convenience.

Supplementary Figure 5. The representative optical images of AA stacking MoS₂ bilayer samples obtained under different growth time of C-D stage at 750 °C (for the second layer growth): (a) 5 min, (b) 10 min, (c) 15 min, and (d) 20 min.

Comment 3. Why is the TMD layer not grown on SiO₂ during the second layer growth?

Response/corrections: We thank the Reviewer for this insightful comment. This is because the second monolayer crystals prefer to grow epitaxially and homogeneously on the activated nucleation centers of the first monolayer. We have systematically studied the structure evolution of the as-grown AA stacking bilayer samples with the growth time of C-D stage for the second layer growth, and the representative results are shown in Figure S5 (see Comment 2 above). One can observe some individual monolayer triangles on the growth substrate especially when the growth time of the second layer is less than 12 minutes like 5 and 10 minutes (Supplementary Figure 5a and 5b). However, when the growth time of the second layer is long enough like 15 and 20 minutes, the complete AA stacking bilayer samples can be predominately synthesized on the whole substrate surface without any monolayer triangles (Supplementary Figure 5c and 5d).

Therefore, it is more reasonable to attribute the individual monolayer triangles on the SiO₂ substrate to the first layer growth since long time growth (exceeding 12 minutes) of the second layer could fully cover the surface of the first monolayer and result in predominantly bilayer samples without any monolayer triangles. Why the TMD layer does not grow on the SiO₂ substrate during the second layer growth is explained on page 5 of the revised manuscript (marked by red color). This discussion is copied below for the Reviewer's convenience. References are numbered as in the revised manuscript.

The strikingly positive effect of such reverse N₂/H₂ flow during the temperature swing stage can be explained as follows. First, the reverse carrier gas flow can prevent unintended supply of chemical vapor source to eliminate the generation of new nucleation centers on the growth substrate and the as-grown first layer during the temperature rising process⁴⁰. Zhang et al.⁸ employed the same idea to prevent uncontrolled homogeneous nucleation and thus enabled highly robust epitaxial growth of diverse heterostructures, multi-heterostructures and superlattices from 2D atomic crystals. Second, the hydrogen flow could saturate the dangling bonds on the edge and at the surface of the as-grown first MoS₂ monolayer crystals, thus blocking the laterally epitaxial growth as reported by Jia et al⁴¹. The surface energy of the edge-terminated structure is considerably higher compared to that of the as-grown flat basal-plane structure^{42,43}. Consequently, the second monolayer is more likely to deposit on the as-grown monolayer surface. At the same time, H₂ can slightly etch away emerging nucleation points on the growth substrate and thereby reduce the wettability of the growth substrate⁴⁴. As such, the source vapor during the C-D stage can be easier to transfer through the substrate surface and have more possibility to reach the surface of the as-grown first monolayer. With a suitable carrier gas flow rate, the source vapor can have enough kinetic energy to reach the surface center of the as-grown first monolayer where there is an initial nucleation to begin the growth of the second layer⁴⁵. Therefore, the second monolayer crystals prefer to grow epitaxially and homogeneously on the activated nucleation centers of the first monolayer, finally promoting the growth of MoS₂ bilayer crystals, as shown in Fig. 1b.

Comment 4. *If the reverse flow is deciding the stake type then grow third monolayer and show the AAA and ABA stakes growth.*

Response/corrections: We thank the Reviewer greatly for this constructive comment. Indeed, our method is suitable to grow trilayer and even thicker structures. Following the

Reviewer's suggestion we have included some results on the growth of trilayer structures. However, as the focus of this work is specifically on bilayer structures and the unique properties associated with them, we have placed these results to Supporting Information. The growth parameters (including the selectivity between the AAA and AAB stackings as well as between the ABA and ABB stackings) at this stage are not optimised and it will be the subject of future work. Nonetheless, we have included a sentence (See Page 8 of the revised manuscript) and a detailed discussion (See Pages 8 and 9 of SI) where we provide practical insights on how to achieve the required trilayer stacking using our growth method.

As we emphasized in the response to Comment 3, the reverse flow can prevent unintended supply of chemical vapour source to eliminate the generation of new nucleation centres on the growth substrate and the as-grown first layer and thus promote the epitaxial growth of the second monolayer, but is not the only deciding factor of the different stacking orders. In fact, the AA and AB stacking orders are decided by their stability and thermodynamic energy. Both the AA (with the twist angle $\theta=0^{\circ}$) and AB (with the twist angle $\theta=60^{\circ}$) stacking MoS₂ bilayers are the most stable structures, and this has already been proved [Nanoscale 2017, 9, 13060; J. Phys. Chem. C 2014, 118, 9203; ACS Nano 2015, 9, 12246]. The stability of the bilayer crystals is related to the formation energies, defined as the total energy difference per atom between the bilayer and the two constituent but separated single monolayers ($E_{\text{form-2L}} = E_{\text{bilayer}} - 2E_{\text{monolayer}}$). The larger is the absolute value of the formation energies, the more stable is the growing structure. Therefore, the growth of every monolayer is sensitive to the growth parameters, especially the growth temperature difference between the two adjacent monolayers and the gas flow rate during the growth process.

We used our reverse-flow epitaxial growth method to grow MoS₂ trilayers and the preliminary results with AAA, AAB, ABB and ABA stacking orders are shown in Supplementary Figure 7. By using reverse-flow method and setting appropriate temperature steps and gas flow rates for the third layer growth, we obtained large-area and large-size AAA and AAB stacking trilayer crystals as clearly reflected by the optical images (Supplementary Figures 7a, 7b and 7d) and the AFM surface morphology image (Supplementary Figure 7c). However, those MoS₂ trilayer crystals with other stacking orders like ABB and ABA can be occasionally obtained and usually coexist on the same substrate. The formation and coexistence of MoS₂ trilayer crystals with different stacking may be related to their formation energies and their stability. Similar to the case of bilayer crystals, the formation energies of trilayer crystals can

be defined as the total energy difference per atom between the trilayer and both the bilayer and monolayer ($E_{\text{form-3L}} = E_{\text{trilayer}} - E_{\text{monolayer}} - E_{\text{bilayer}}$). The growth parameters (including the selectivity between the AAA and AAB stacks as well as between the ABA and ABB stacks) at this stage are not optimised and it will be the subject of future work.

Supplementary Figure 7. The optical images of trilayer MoS₂ with AAA stacking order; (c) the AFM surface morphologies of the trilayer MoS₂ with AAA stacking order; the optical images of the trilayer MoS₂ with (d) AAB, (e) ABB and (f) ABA stacking order; (g)-(h) the optical images of multilayer MoS₂.

Comment 5. *There is possibility in AB stake (high temp. growth) the stress is different from AA stake (low temp. growth) and which could be a deciding factor for materials property. What is your comment on stress?*

Response/corrections: We thank the Reviewer very much for pointing out on this possibility. We have carefully considered the growth conditions for AB and AA stacks, and concluded that stress plays a less significant role in this case. This is because the growth temperature difference between AA and AB samples is only 50 °C, which should not induce

large stress difference between them. Furthermore, the tube system is slowly and naturally cooled down to room temperature after the second layer growth, so the stress for both AA and AB samples may have been released to a certain extent. Both Raman and PL spectra have previously been shown to be quite sensitive to the stress [Sci. Rep. 2014, 4, 5649; Nano Lett. 2013, 13, 3626]. Importantly, our results performed on the AA and AB stacking MoS₂ bilayer crystals (Fig. 3a and 3b) did not show any obvious difference between them, further proving that the stress difference induced by the growth temperature difference does not play a significant role in our case.

Comment 6. *Author has grown big flakes but AFM, Raman and PL measurements were done on small flakes, why? are the properties of small and big flakes same? Provide some evidence.*

Response/corrections: We thank the Reviewer for this comment. The small flakes were chosen for AFM measurements because AFM can measure the whole flakes so that we can easily distinguish the stacking order from the AFM pictures of the whole flakes. The other techniques including Raman and PL mappings were performed on the same samples for consistency. We did perform AFM measurements on big flakes as shown in Figure 6e. In such case, the AFM image only shows a corner of the big triangle flake. On the other hand, both Raman and PL mapping measurements often need quite long integration time, for example, several hours to scan a flake with tens of microns, so we usually select small flakes for Raman and PL mapping studies.

We believe that the structural characteristics of MoS₂ flakes are indeed not sensitive to the flake size. It has been proved that the uniformity of the layer number can be identified by the optical contrasts between the flakes and the substrates [Nano Lett. 2007, 7, 2758]. The uniform optical contrasts of the optical images given in the whole manuscript strongly prove the uniformity of the as-grown bilayer flakes. Here we also performed Raman and PL mapping studies on two big MoS₂ bilayer flakes (AA and AB stacking) with a size of about 100 μm in Figure S10 (which is copied below for the Reviewer's convenience). The results reflect the same properties of the big flakes as those of the small flakes. Furthermore, both Raman and PL intensity on the boundaries of the big flakes are much clearer than those of the small flakes. The corresponding discussion has been added to Page 11 of Supporting Information.

Supplementary Figure 10. Optical images, Raman and PL intensity maps of two big MoS₂ flakes with a size of about 100 μm: (a) AA stacking order and (b) AB stacking order.

Comment 7. In figure 3(e), PL brightness (intensity) should be higher at the edge of flakes due to single layer but not observed, why? However, it can be clearly seen in fig 3(h) of AB stake.

Response/corrections: We thank the Reviewer for this comment. In Fig.3e, the AA stacking bilayer MoS₂ flake we chose is a complete one with no steps between the upper and the bottom layers. The two adjacent layers have the same size, so it is a complete bilayer crystal which possesses uniform PL intensity distribution. However, the AB stacking bilayer as shown in Fig.3h has quite clear steps between the two layers with the centre area representing a bilayer structure and the edge being the monolayer one. The PL intensity of the monolayer MoS₂ is far stronger than that of the bilayer MoS₂, so it can be clearly distinguished from the PL intensity mapping, as shown in Fig.3h. Here we have also performed PL intensity mapping on the incomplete AA stacking bilayer one with distinct steps in Fig. S9 (which is copied below for the Reviewer's convenience). As seen, there are also clear differences between the monolayer and the bilayer area. The Raman intensity of E_{2g}¹ mode of the bilayer MoS₂ in the centre area is much stronger than that of the monolayer one on the edge, while the PL intensity of the monolayer MoS₂ on the edge is evidently much stronger than that of the bilayer one in the centre area. The corresponding discussion has been added to Page 10 of Supporting Information.

Supplementary Figure 9. (a) optical image, (b) Raman intensity mapping at around 382 cm^{-1} (E^1_{2g} mode) and (c) PL intensity mapping at around 678 nm (A exciton) of an incomplete AA stacking bilayer MoS_2 sample with steps between the two layers.

Comment 8. The I_{ds} - V_{ds} plot is linear for AA and rectifying for AB stakes, why?

Response/corrections: We thank the Reviewer for this insightful comment. The FET devices based on both AA and AB stacking samples were fabricated with the same electrode materials and the same electron beam lithography (EBL) methods. The differences of output characteristic (I_{ds} - V_{ds}) between these two kinds of devices can only be related with their different stacking orders and their different contact arrangements/junctions with the metal electrodes. As shown in Supplementary Figure 14a, both the drain and source electrodes fall on the bilayer area for complete AA stacking bilayer FETs, so uniform bilayer acts as the FET channel. This situation is quite similar to MoS_2 monolayer based FETs. MoS_2 bilayer FETs have more electrons injected into the channel than the monolayer based FETs [Sci. Rep. 2016, 6, 21786], leading to Ohmic contacts as reflected by the linear I_{ds} - V_{ds} plots shown in Figure 5b as well as Figure S11. Similar results with Ohmic contacts between metal electrodes and MoS_2 bilayer have also been reported previously [Nano Lett. 2012, 12, 4674]. In contrast, there are steps between the two layers for AB stacking bilayer FETs, so the drain and source electrodes fall partly on the bilayer area and partly on the monolayer area as shown in Supplementary Figure 14b. Therefore, the electron density in the AB stacking bilayer channel are much lower compared to the AA stacking channel [Sci. Rep. 2016, 6, 21786]. Furthermore, the stacking orders have also been shown to significantly affect the electronic characteristics of 2D materials [J. Phys. Soc. Jpn. 2007, 76, 024701; Nanoscale 2015, 7, 14062; Carbon 1994, 32, 289; Phys. Rev. B 2014, 89, 075409]. Above all, we predict that the AB stacking order together with the uneven electrode distribution contribute

to the Schottky contacts as revealed by the I_{ds} - V_{ds} curves in Figure 5d as well as Figures S12b, S12d and S12f.

We also performed electrical studies on a recently prepared AB stacking bilayer FET device with the similar rectification characteristic and output curves (I_{ds} - V_{ds}) as those presented in Figure 5d and Supplementary Figures 12b, 12d and 12f. The output characteristics were measured by switching the source and drain electrodes (electrode 1 and 2) and the results are shown in Supplementary Figure 14c. It is clear that both output curves are of rectification characteristic with slight inconsistency, meaning that both electrode contacts are slightly asymmetrical and of Schottky type. Similar effects on MoS₂ bilayer FET devices have been systematically studied by Bartolomeo et al. [Adv. Funct. Mater. 2018, 28, 1800657]. As they proved, such two asymmetrical Schottky contacts constitute two back-to-back junctions in the Electrode 1/MoS₂ channel/Electrode 2 configuration, which is thus responsible for the rectifying phenomena of MoS₂ bilayer FET devices including AB stacking samples in this work. Relevant corrections include Fig S14 below and associated discussion in Pages 13 and 14 of the Revised manuscript as well as in Pages 15 and 16 of Supporting Information.

Supplementary Figure 14. The electrode configuration schematics of (a) AA and (b) AB stacking bilayer MoS₂ FET devices; (c) The output characteristics (I_{ds} - V_{ds}) of a representative AB stacking bilayer MoS₂ FET device measured by switching source and drain electrodes (between electrode 1 and 2). Inset in (c) is the optical image of the AB stacking bilayer MoS₂ FET device.

Comment 9. The conduction is through top layers so I-V behaviour should be same. Author should provide a clear conduction mechanism.

Response/corrections: We thank the Reviewer for this comment. As we emphasized in the response of Comment 8, the differences between these two kinds of devices can only be

related with their different stacking orders and their different contact arrangements/junctions with metal electrodes. First, it has been demonstrated that the stacking orders have great impact on 2D materials' electronic characteristics [J. Phys. Soc. Jpn. 2007, 76, 024701; Nanoscale 2015, 7, 14062; Carbon 1994, 32, 289; Phys. Rev. B 2014, 89, 075409]. Second, the AB stacking bilayer MoS₂ samples have large steps between the two layers, so the electrodes locate partly on the bilayer and partly on the monolayer, which is quite different from the situation of the AA stacking bilayer FET devices as shown in Supplementary Figures 14a and 14b. It is these two differences that lead to the different output characteristics as observed for AA and AB stacking bilayer FET devices. The discussion of the conduction mechanisms has been incorporated in the revisions arising from our response to Comment 8.

Comment 10. *Compare the I_{DS} - V_{DS} behaviour of mono and bi layers.*

Response/corrections: We thank the Reviewer for this comment. We have fabricated FETs on the as-grown monolayer MoS₂ triangles by conventional CVD method for comparison and presented a set of representative results in Figure S13 (which is copied below for the Reviewer's convenience). The output curves (I_{DS} - V_{DS}) show an extremely slight rectification characteristic, indicating the extremely slight Schottky contact between the electrode and monolayer. Similar results have also been reported previously [Appl. Phys. Lett. 2013, 102, 193107]. In fact, whether for monolayer or bilayer MoS₂ FETs, the contacts can be of Ohmic and Schottky types as proved by many literatures [Nat. Nanotechnol. 2011, 6, 147; Appl. Phys. Lett. 2013, 102, 193107; ACS Nano 2014, 8, 5633]. The results varied from devices to devices. Just take the case of AA stacking bilayer MoS₂ FETs in this work, some are of Ohmic type as shown in Figure 5b and S11d, and some are of extremely slight Schottky type as shown in Figure S11b and S11f. However, for AB stacking ones in this work, all are of Schottky type. There are a lot of factors such as the work function of the electrode material, the layer number of MoS₂ and the unavoidable defects formed during the fabrication process that can influence the output characteristic [Adv. Funct. Mater. 2018, 28, 1800657; ACS Nano 2014, 8, 2880]. A short discussion on the I_{DS} - V_{DS} characteristics of monolayer and bilayer based devices can be found in Page 14 of Supporting Information.

Supplementary Figure 13. The output characteristics (I_{ds} - V_{ds}) at varying V_g of two FETs based on conventional CVD-grown monolayer MoS_2 .

Comment 11. Why mobility is low in AB stakes? The mobility should be same.

Response/corrections: We thank the Reviewer for this comment. For one hand, as shown in Figures 5, S11 and S12, the FETs based on AA stacking bilayer samples possess Ohmic contact characteristics, while those based on AB stacking bilayer samples have Schottky contact characteristics. The sizeable Schottky barriers for AB stacking based FETs may limit the current output and thus lead to lower, extrinsic mobility values since the electrical response was measured in a two contact configuration as previously reported [Appl. Phys. Lett. 2013, 102, 123105]. On the other hand, the contact junction between the electrode and the surface is quite different for these two kinds of devices as we addressed in the response of Comment 8. For AA stacking ones, the electrodes fall completely on the bilayer surface, while for AB stacking ones, the electrodes fall partly on the bilayer surface and partly on the monolayer surface. As such, the electron density in the AB stacking bilayer channel are much

lower than that of the AA stacking channel [Sci. Rep. 2016, 6, 21786], resulting in the lower carrier mobility.

In addition, it has already been proven that the energy band structures are highly related to the stacking order of bilayer or few-layer 2D materials due to their different interlayer interaction between the layers [Phys. Rev. B 2014, 89, 075409; Nano Lett. 2015, 15, 8155; J. Phys. Soc. Jpn. 2007, 76, 024701; Carbon 2011, 50, 784]. The different energy band structures between AA and AB stacking (Figure 4) may also result in some different mobility values. This point was emphasized in Page 14 of the revised manuscript.

Comment 12. *Could you control the value of x and y in $Mo_{1-x}W_xS_2(1-y)Se_y$?*

Response/corrections: We thank the Reviewer greatly for this comment. The quarternary $Mo_{1-x}W_xS_2(1-y)Se_y$ bilayer compounds were just introduced as a proof of concept that quarternary bilayer crystals can also be produced with a reasonable quality by our reverse-flow chemical vapour epitaxy. However, controlling the value of x and y is out of the scope of this work. In previous reports [Nanoscale 2015, 7, 13554; ACS Catal. 2015, 5, 2213], the values of x in the ternary alloys such as $Mo_{(1-x)}W_xS_2$ and $MoS_2(1-x)Se_{2x}$ are usually tuned by adjusting the ratio of these source powders. Naturally, the values of x and y in $Mo_{1-x}W_xS_2(1-y)Se_y$ may also be controlled by tuning the ratio of these source powders. This study will be a subject of our future work. Until now, we have achieved some preliminary results by fixing MoO_3 , S and Se powder amounts and varying WO_3 powder amount, as shown in Supplementary Figure 16 (which is copied below for the Reviewer's convenience). Raman and PL spectroscopies are powerful and nondestructive characterization tools to determine the kinds of TMD materials [Phys. Rev. B 2013, 88, 245403; ACS Nano 2014, 8, 9649] and their elemental composition [J. Mater. Chem. C 2015, 3, 2589; Nano Lett. 2015, 15, 8155]. As can be seen from Supplementary Figure 16a, with the amount of WO_3 increasing, the WS_2 -related Raman peaks become more and more obvious and have a little redshift. At the same time, the MoS_2 -related E_{2g}^1 modes are gradually red-shifted while the MoS_2 -related A_{1g} modes are gradually blue-shifted. On the other hand, as shown in Supplementary Figure 16b, with the amount of WO_3 increasing, the three PL peaks of the quarternary bilayer crystals all have some red shift. Both the Raman and PL results verify that the value of x and y in Mo_{1-x}

$xW_xS_{2(1-y)}Se_y$ can be controlled by our reverse-flow chemical vapour epitaxy method. Relevant contents can be found in Pages 17 and 18 of SI.

Supplementary Figure 16. (a) Raman and (b) PL spectra of the $Mo_{1-x}W_xS_{2(1-y)}Se_y$ bilayer alloys obtained by fixing MoO_3 , S and Se powder amounts and varying WO_3 powder amount using our reverse-flow chemical vapour epitaxy method.

Response to the comments and suggestions of Reviewer #3

General comment: *This is a very interesting work about the growth of transition metal dichalcogenides bilayer single crystals using a method of reverse flow chemical vapor epitaxy. A series of TMD bilayer crystals including MoS₂ and WS₂, ternary Mo(1-x)WxS₂ and quaternary Mo(1-x)WxS₂(1-y)Se_y are synthesized with variable structural configurations and tunable electronic and optical properties. In addition, TMDC bilayer crystals with AA and AB stacking structures can be obtained by adjusting the growth temperature of the second monolayer. These results representing a promising TMDC materials system for fundamental studies and technological applications. The article is suggested to be publish after revised.*

Response/corrections: We are grateful for the positive comments given by the Reviewer. We have made substantial improvements in the revised manuscript according to the Reviewer's comments and hope that the manuscript is now suitable for publication.

Comment 1. *Can you show more information about the growth under different temperatures (e.g., 720 °C, 760 °C, etc.)? In this way, the relationship between temperature and the epitaxial growth behaviour of the second monolayer can be demonstrated more clearly.*

Response/corrections: We thank the Reviewer for this insightful comment. We did grow MoS₂ bilayer crystals by setting different growing temperatures of C-D stage (e.g., 720 °C, 770 °C and 790 °C) for the second layer growth and the results are presented in Supplementary Figure 4. For 720 °C which is quite close to the growing temperature of the first monolayer (A-B stage), one can still observe MoS₂ monolayer crystals dominating over the resultant flakes on the substrate surface no matter how we prolong the growth time of C-D stage (Supplementary Figure 4b). For 770 °C, the experimental result of crystal growth is quite similar to that obtained at 750 °C as presented in Supplementary Figure 4d, where uniform AA stacking MoS₂ bilayer crystals are largely distributed on the substrate surface. For 790 °C, however, both AA and AB stacking MoS₂ bilayer crystals coexist on the same substrate surface as shown in Supplementary Figure 4e. Until 800 °C, AB stacking MoS₂ bilayer crystals become dominant over the resultant flakes on the substrate surface.

As we emphasized, the reverse flow can prevent unintended supply of chemical vapour source to eliminate the generation of new nucleation centres on the growth substrate and

the as-grown first layer and thus promote the epitaxial growth of second monolayer, but is not the only deciding factor of the different stacking orders. In fact, the AA and AB stacking orders are decided by their stability and thermodynamic energy. Both the AA (with the twist angle $\theta=0^\circ$) and AB (with the twist angle $\theta=60^\circ$) stacking MoS₂ bilayers are the most stable structures, and this has already been proved [Nanoscale 2017, 9, 13060; J. Phys. Chem. C 2014, 118, 9203; ACS Nano 2015, 9, 12246]. The stability of the bilayer crystals is related to the formation energies, defined as the total energy difference per atom between bilayer and the two constituent but separated single monolayers ($E_{\text{form-2L}} = E_{\text{bilayer}} - 2E_{\text{monolayer}}$). The larger the absolute value of the formation energies, the more stable the growing structure. Therefore, the growth of every monolayer is sensitive to the growth parameters, especially the growth temperature difference between the two adjacent monolayers and the gas flow rate during the growing process.

Supplementary Figure 4. The representative optical images of MoS₂ crystals obtained under different growing temperatures of C-D stage: (a) 700 °C, (b) 720 °C, (c) 750 °C, (d) 770 °C, (e) 790 °C and (f) 800 °C.

The growth results under different temperature (e.g., 700 °C, 720 °C, 750 °C, 770 °C, 790 °C and 800 °C) of C-D stage suggest that the growth temperature difference between the two adjacent monolayers together with the reverse flow determines the stacking order of such bilayer growth. It has ever been reported that bare SiO₂/Si is more wettable and has higher total surface energy when compared to MoS₂/SiO₂/Si [Nanoscale 2016, 8, 5764]. In addition, the increasing growing temperature can reduce the wettability of the growth substrate

[Nanoscale 2016, 8, 5764]. For 720 °C or even lower, the increasing growing temperature between A-B stage and C-D stage is not large enough to reach a low enough wettability so that vertical epitaxy is suppressed leading to monolayer crystals dominating over the resultant flakes. For 750 °C or even higher, the increasing growing temperature is large enough which can effectively suppress the new nucleation on the substrate and lead to the vertically epitaxial growth on the as-grown monolayer MoS₂ [J. Am. Chem. Soc. 2015, 137, 14281]. For 750~780 °C, the growth temperature difference between the two adjacent monolayers is suitable for the formation of AA stacking. For 800 °C or even higher, the growth temperature difference between the two adjacent monolayers is optimum for the formation of AB stacking.

We have placed these results and the corresponding discussion in Pages 4-6 of Supporting Information.

Comment 2. *It's necessary to show relationship between the size of the second layer MoS₂ and the growth time at 750°C? This will gives a more clearer picture of the growth process.*

Response/corrections: We thank the Reviewer for this useful comment. We have to admit that it is hard to precisely and quantitatively study the size of the second layer MoS₂ with the growth time. First, the size of the second layer is bound by the first layer, in other words, the edge of the first layer acts as a limiting factor in the growth of the second layer. It is difficult to overcome the potential barrier of the steps between the first layer and the growth substrate. Second, the size of the monolayer MoS₂ grown at stage A-B usually varies from sample to sample leading to the different size distribution of the resultant bilayer crystals. Third, it couldn't be confirmed that all the second layers start the growth on the as-grown monolayers simultaneously so that the size of the second layer can change from flake to flake even on a same growth substrate.

Nevertheless, we can qualitatively study the size of the second layer MoS₂ with the growth time at 750 °C and the results are shown in Figure S5. The corresponding discussion together with Figure S5 is attached below for the Reviewer's convenience and can be referred to the paragraph marked by red color in Page 7 of the revised manuscript as well as Pages 6 and 7 of SI. Reference numbering is the same as in the revised manuscript.

Interestingly, the structure of the as-grown AA stacking bilayer MoS₂ samples displays a close relationship with the growth time of C-D stage (for the second layer growth), as presented in the optical images of the as-grown AA stacking bilayer MoS₂ samples under different growth time of C-D stage (Figure S5). When the growth time of C-D stage was shortened to less than 10 minutes (like 5 minutes), the epitaxial growth of the second monolayer is insufficient to fully cover the surface of the first monolayer so that the incomplete AA stacking bilayer MoS₂ grains with distinct steps can be achieved as shown in Fig. 2c and Fig. S5a. The corresponding AFM image in Fig. 2d explicitly display the step between the substrate surface and the first monolayer as well as that between the first monolayer and the second monolayer, both of which are characteristic of MoS₂ monolayer⁴⁶. However, when the growth time was kept for about 10 minutes, both complete and incomplete AA stacking samples can coexist on the substrate surface as presented in Fig. S5b. When the growth time of C-D stage was kept for more than 12 minutes (like 15 and 20 minutes), the complete AA stacking bilayer samples without any steps can be predominately synthesized on the whole substrate surface as observed in Fig. 2a, Figs. S5c and S5d.

Supplementary Figure 5. The representative optical images of AA stacking MoS₂ bilayer samples obtained under different growth time of C-D stage at 750 °C (for the second layer growth): (a) 5 min, (b) 10 min, (c) 15 min, and (d) 20 min.

Comment 3. *As we can see in the Fig 2(e), the second layer of A-B stacking MoS₂ cannot completely cover the first layer . What's the reason? Can you get a full coverage A-B stacking MoS₂?*

Response/corrections: We thank the Reviewer for this comment. The second layer of A-B stacking MoS₂ cannot completely cover the first layer because the second layer will stop growing when it reaches the edge of the first layer. It is difficult to overcome the potential barrier of the steps between the first layer and the growth substrate. This is a common effect for CVD-grown AB stacking MoS₂ bilayer crystals, whether in this work (Figure 2e and 2f) or in the previous reports [ACS Nano 2015, 9, 12246; Nat. Commun. 2014, 5, 4966]. It is the same for CVD-grown AA stacking MoS₂ bilayer crystals. The second layer will stop growing when it reaches the edge of the first layer, resulting in complete AA stacking ones as shown in Figure 2a and 2b. Furthermore, the second layer may stop growing before reaching the edge of the first layer, resulting in incomplete AA stacking ones with steps as shown in Figure 2c and 2d. Relevant short discussion can be found in Pages 6 and 7 of SI.

Comment 4. *In Fig. 2(i), In addition to the normal white diffraction spots, we can also find some unnormal small points that are not obvious. What's the reason?*

Response/corrections: We thank the Reviewer for this comment. We doubt that these abnormal small points may be due to instrument interference for the following reasons. First, the growth process was extra clean and no other impurities were introduced as proved by Raman, PL and XPS results. Second, these small points cannot form any independent diffraction patterns as an indication of other monocrystalline or polycrystalline materials. Even so, the clear and bright FFT patterns in the original Fig. 2i clearly reveal the hexagonal symmetry lattice structures of MoS₂ bilayer crystals. We also changed the position to redo the FFT pattern and the result is shown in the revised Fig. 2i. No such interference spots can be observed anymore. We here also copy the revised Fig. 2 below for the Reviewer's convenience.

Fig.2 A representative optical microscope image (a) and a representative AFM surface morphology image (b) of the as-grown AA stacking bilayer MoS₂ samples with the maximum lateral size of even up to 300 μm; a representative optical image (c) and a representative AFM surface morphology picture (d) of the AA stacking bilayer MoS₂ grains with distinct steps, where the twist angle from the relative rotation of the two vertically stacked triangles $\theta=0^\circ$ indicate the AA stacking order^{37,50,51}; a typical optical image (e) and a typical AFM surface morphology image (f) of the as-grown AB stacking bilayer MoS₂ samples, where the twist angle from the relative rotation of the two vertically stacked triangles $\theta=60^\circ$ indicate the AB stacking order^{37,50,51}; planar TEM images of the AA stacking bilayer MoS₂ crystals: low resolution (g) and high resolution (h) as well as the selected area electron diffraction (SAED) patterns (i); the inset in (g) shows the folded edge of MoS₂ bilayer films.

Comment 5. Can you get trilayer or more layer 2D-TMD through this method?

Response/corrections: We thank the Reviewer greatly for this constructive comment. Indeed, our method is suitable to grow trilayer and even thicker structures. Following the Reviewer's suggestion we have included some results on the growth of trilayer and even multilayer structures. However, as the focus of this work is specifically on bilayer structures and the unique properties associated with them, we have placed these results to Supporting Information. The growth parameters (including the selectivity between the AAA and AAB

stackings as well as between the ABA and ABB stackings) at this stage are not optimised and it will be the subject of future work. Nonetheless, we have included a sentence (See Page 8 of the revised manuscript) and a detailed discussion (See Pages 7 and 8 of SI) where we provide practical insights on how to achieve the required trilayer or even multilayer stackings using our growth method. The newly added Fig. S7 showing various trilayer structures and multilayer structures is also shown below for the Reviewer's convenience.

Supplementary Figure 7. The optical images of trilayer MoS₂ with AAA stacking order; (c) the AFM surface morphologies of the trilayer MoS₂ with AAA stacking order; the optical images of the trilayer MoS₂ with (d) AAB, (e) ABB and (f) ABA stacking order; (g)-(h) the optical images of multilayer MoS₂.

Response to the comments and suggestions of Reviewer #4

General comment: *The authors present a detailed study and description of a novel experimental procedure based on reverse-flow chemical vapor epitaxy which allows them to produce large-size TMD bilayer crystals of high quality. The quality of the samples produced via this method is then tested and the samples are characterized by a variety of methods and techniques, indicating the successful synthesis of high quality crystals. In addition, the authors have performed electrical transport studies which show that the bilayer MoS₂ samples have better electronic performance than the monolayer counterparts. To support their experimental observations, the authors have performed DFT calculations, from which they have generated band structure plots for bilayer MoS₂ samples with AA and AB stacking. Although their theoretical model is simplified, it seems to be sufficient to support their experimental PL findings. Since they are studying bilayer structures, however, it is important to include van der Waals interactions, which have been ignored in this model system. In addition, the authors should present additional details on the theoretical methods which are needed to reproduce the model. For example, they need to report the lattice constant for MoS₂, and details about force convergence. There is also a typo in the description of the PBE functional, it should read "Perdew-Burke-Ernzerhof".*

Overall the manuscript has high value as far as the experimental method that is presented here is concerned. The authors claim that the method can be universally applied to produce bilayers of different TMD compounds, which is valuable for the design of advanced electronics. The theory part is simple, and the bilayer model of TMDs has been extensively studied from other groups. Therefore, it doesn't add much value to the article. The authors could have used band structure plots that are available in the literature to support their experimental observations.

Response/corrections: We are grateful for the positive comments given by the Reviewer. We agree with the Reviewer that our theory calculations are simple, but we used them to support the PL characteristics and stacking orders of both AA and AB stacking MoS₂ bilayer crystals. We did include van der Waals interactions in this model system although we didn't describe these interactions in the method part of the original manuscript, for which we do apologize. A DFT-D2 dispersion correction method of Grimme is used to describe van der Waals interactions in all models. We have made a more detailed description on the theoretical methods we used in the revised manuscript according to the Reviewers' comments. In addition, we have correct the typo to be "Perdew-Burke-Ernzerhof". Please refer to the sentences marked by red colour in Pages 20 and 21 of the revise manuscript.

The paragraph regarding the theoretical methods is also copied below for the Reviewer's convenience.

The geometry optimizations and the energy band structure calculations of the three kinds of MoS₂ models, the monolayer one, the bilayer ones with AA and AB stacking orders, were calculated through First-principles based on density functional theory (DFT), which were implemented in the Atomistic-ToolKit (ATK) version 2017.0. The Perdew-Burke-Ernzerhof (PBE) of generalized gradient approximation (GGA) was used for studying the band structures. Besides, pseudopotentials of MoS₂ using the Hartwingster-Goedecker-Hutter scheme with Tier 4 basis set. A DFT-D2 dispersion correction method of Grimme was used to describe van der Waals interactions in all models. The density mesh cutoff was set as 75 Hartree to achieve the energy convergence of 10⁻⁵ eV and the force convergence of 0.01 eV/Å on each atom. The Brillouin zone was sampled by 9×9×1 and 25×25×1 Monkhorst-Pack K-point mesh for structure optimization and electronic band calculation. A vacuum spacing of 15 Å was used to avoid interaction between the calculated individual structures. The optimized lattice constant of MoS₂ by our PBE-GGA calculation is 3.16 Å.

Comment 1. *Figure 1 has (a), (b) and (c) parts, but the authors refer to Fig. 1d on page 5, line 123 and page 6, line 138.*

Response/corrections: We thank the Reviewer for pointing out this error. We have corrected Fig. 1d as Fig. 1c both in the caption of Fig. 1 and in Page 7 of the revised manuscript.

Reviewers' comments:

Reviewer #1 (Remarks to the Author):

The points raised in the previous round of review have been satisfactorily addressed. The manuscript can be published as it is.

Reviewer #2 (Remarks to the Author):

Author has incorporated all the queries.

Reviewer #3 (Remarks to the Author):

The author give good answers for the questions. And the paper recommended to be published after correcting some minor mistakes ,such as: the description of SEAD in Figure 2 should be changed to FFT.

Reviewer #5 (Remarks to the Author):

The manuscript presents an improved experimental procedure of the thermal CVD process based on reverse-flow chemical vapor epitaxy so that large size TMD bilayer crystal can be grown with high quality. The DFT calculation results are also presented in order to support the experimental findings. A few concerns are raised regarding to the DFT results presented in the manuscript.

- (1) The vdw interaction is very important to describe the inter-layer interaction correctly of the bi-layer MoS₂ with AB/AA stacking. DFT-D2 dispersion correction is one of the method to add the vdw interactions but other methods are available as well, such as DFT-TS correction (A. Tkatchenko and M. Scheffler, 'Accurate molecular van der waals interactions from ground-state electron density and free-atom reference data', Phys. Rev. Lett. 102, 073005 (2009)). However, both of these correction method are developed based on other material system. With these vdw correction added, the lattice parameters will be modified accordingly. The authors are recommended to list the reason why DFT-D2 correction is selected and can be adopted in this study. The authors are recommended to benchmark their monolayer MoS₂ results with other available publications, in terms of energy, lattice parameters (bond length, bond angle, interlayer distance, etc).
- (2) In Fig 4 on page 12, (a), (b), (c) are missing.
- (3) The band structure energy range (y axis) is too narrow in Fig 4(a), (b) and (c). It is hard to compare the authors results with other published DFT work.
- (4) The authors are recommended to adopt DFT calculation to answer the following questions in order to support their experimental findings: such as (a) why AA/AB stacking are preferred. Which one is more energetic favorable. (b) what causes the triangle (hexagonal) shape of the CVD grown MoS₂? (c) if more than bi-layer are grown, which stacking is energetic favorable. (d) what is the reason behind triangle orientation change in Fig S7(h).

Response to the comments and suggestions of Reviewer #3

General comment: The author give good answers for the questions. And the paper recommended to be published after correcting some minor mistakes ,such as:the description of SEAD in Figure 2 should be changed to FFT.

Response/corrections: We thank the Reviewer for the positive comment. We have changed the description of SEAD in Figure 2 to FFT. Please refer to Figure 2 caption.

Response to the comments and suggestions of Reviewer #5

General comment: *The manuscript presents an improved experimental procedure of the thermal CVD process based on reverse-flow chemical vapor epitaxy so that large size TMD bilayer crystal can be grown with high quality. The DFT calculation results are also presented in order to support the experimental findings. A few concerns are raised regarding to the DFT results presented in the manuscript.*

Response/corrections: We thank the Reviewer for these positive comments. We have made a substantial improvement in the revised manuscript according to the Reviewers' comments and hope that the manuscript is now suitable for publication.

Comment 1. *The vdw interaction is very important to describe the inter-layer interaction correctly of the bi-layer MoS₂ with AB/AA stacking. DFT-D2 dispersion correction is one of the method to add the vdw interactions but other methods are available as well, such as DFT-TS correction (A. Tkatchenko and M. Scheffler, 'Accurate molecular van der waals interactions from ground-state electron density and free-atom reference data', Phys. Rev. Lett. 102, 073005 (2009)). However, both of these correction method are developed based on other material system. With these vdw correction added, the lattice parameters will be modified accordingly. The authors are recommended to list the reason why DFT-D2 correction is selected and can be adopted in this study. The authors are recommended to benchmark their monolayer MoS₂ results with other available publications, in terms of energy, lattice parameters (bond length, bond angle, interlayer distance, etc).*

Response/corrections: We thank the Reviewer for these helpful comments and fully agree that the van der Waals correction is important for studying layered materials. We have referred to this reference [Phys. Rev. Lett. 2009, 102, 073005] when introducing DFT calculation in the Method part of the revised manuscript. The reason why the Grimme's DFT-D2 dispersion-correction approach is selected and adopted in this study is that it can be applicable for all exchange correlation energies due to its higher accuracy and less empiricism [J. Chem. Phys. 2010, 132, 154104]. In a recent paper [Phys. Rev. Mater. 2018, 2, 034005], the authors compared all the van der Waals methods and demonstrated that DFT-D method can work very well for layered materials. Furthermore, such approach has been widely adopted in the calculation of van der Waals interaction system such as transition-metal dichalcogenides (TMDs) [Phys. Rev. B 2012, 86, 075454] and could yield

very good interlayer distances [Phys. Rev. B 2014, 89, 075409], which are proved to have significant effect on the band structures [J. Phys.: Condens. Matter 2014, 26, 405302].

We also performed theoretical calculations using DFT-D3 correction mode to study the energy band diagrams for MoS₂ monolayer, AA stacking MoS₂ bilayer and AB stacking MoS₂ bilayer, and the results are presented in Figure R1. The detailed comparison in terms of lattice constant (A), Mo-S bond length (L), S-Mo-S bond angle (θ) and band gap (E_g) between DFT-D2 and DFT-D3 model is shown in Table R1. As seen, the theoretical results are almost the same for DFT-D2 and DFT-D3 methods, further proving the effectiveness and accuracy of DFT-D2 method for van der Waals interaction.

Figure R1. The calculated energy band structures of MoS₂ monolayer (a), AA-stacking MoS₂ bilayer (b) and AB stacking MoS₂ bilayer (c) using DFT-D3 correction method.

Table R1. Detailed Comparison in terms of lattice constant (A), Mo-S bond length (L), S-Mo-S bond angle (θ) and band gap (E_g) for MoS₂ monolayer, AA stacking bilayer and AB stacking MoS₂ bilayer between DFT-D2 and DFT-D3 model.

Method	MoS ₂	A (Å)	L (Å)	θ (°)	E_g (eV)
DFT-D2	AB	3.161	2.41	81.78	1.24
	AA	3.161	2.41	81.75	1.20
	monolayer	3.161	2.41	81.78	1.79
DFT-D3	AB	3.160	2.41	81.75	1.19
	AA	3.160	2.41	81.77	1.15
	monolayer	3.160	2.42	81.75	1.79

We also benchmark our calculated results of monolayer MoS₂ with other available publications in terms of lattice constant, bond distance and bond angle. The calculated lattice constant of monolayer MoS₂ is 3.16 Å, which is in good agreement with the previous

calculation and experiment results [Phys. Rev. Lett. 2012, 108, 196802; Phys. Rev. B 2011, 84, 153402]. The optimized bond distance between Mo and S atoms is 2.41 Å, and the angle between Mo–S bonds is 81.78°, both of which agree well with the previously reported theoretical results [J. Phys. Chem. C 2014, 118, 1515]. The calculated results of monolayer MoS₂ show that a direct bandgap of 1.79 eV is located at the K point, matching well with the experimental data [Phys. Rev. Lett. 2010, 105, 136805].

In summary, the DFT-D2 correction has been acknowledged as an effective method for the theoretical calculation of VdW interaction schemes. Furthermore, our calculation results are in good agreement with the previously reported theoretical results and experimental data. In fact, the purpose of our calculation using DFT-D2 correction model in this study is to explain the PL behaviors of both as-grown monolayer and bilayer MoS₂. The results show that monolayer MoS₂ has a direct band gap while bilayer one has an indirect band gap, and these are in good agreement with the previous DFT calculations [Nano Lett. 2010, 10, 1271; Small 2014, 10, 1090; Phys. Rev. Lett. 2010, 105, 136805] and experimental results [Nano Lett. 2013, 13, 3626; Nano Lett. 2013, 13, 4212].

We have incorporated this discussion into Supplementary Information (SI). Please refer to the paragraphs marked by red color on Pages 13-14 of SI. Several sentences regarding this discussion has also been added in the revised manuscript (marked by red color on Page 12).

Comment 2. *In Fig 4 on page 12, (a), (b), (c) are missing.*

Response/corrections: We thank the Reviewer for his keen observations. We have labelled (a), (b) and (c) in Fig 4, which is copied below for the Reviewer's convenience.

Fig.4 Atomic structure and band structure of the AA and AB stacking bilayer MoS₂. The calculated energy band structures of MoS₂ monolayer (a), AA-stacking MoS₂ bilayer (b) and AB stacking MoS₂ bilayer (c). The side view (d) and top view in ball-and-stick model (e) as well as top view in MITSUBISHI column model (f) of the atomic structures for AA stacking MoS₂ bilayer; the counterpart views (g), (h) and (i) for AB stacking MoS₂ bilayer. In (d), (e), (g) and (h), the blue solid spheres represent Mo atoms and the yellow ones S atoms. In (f) and (i), the red and blue triangles stand for the bottom and upper MoS₂ layer respectively, where Mo atoms locate at the center of each triangle and S atoms at the apex of each triangle.

Comment 3. The band structure energy range (y axis) is too narrow in Fig 4(a), (b) and (c). It is hard to compare the authors results with other published DFT work.

Response/corrections: We thank the Reviewer for this helpful comment. We have enlarged the energy range (y axis) to -4~4 eV in Fig 4(a), (b) and (c). Please refer to the above Fig. 4 for convenience.

Comment 4. The authors are recommended to adopt DFT calculation to answer the following questions in order to support their experimental findings: such as (a) why AA/AB stacking are preferred. Which one is more energetic favorable. (b) what causes the triangle (hexagonal) shape of the CVD grown MoS₂? (c) if more than bi-layer are grown, which stacking is energetic favorable. (d) what is the reason behind triangle orientation change in Fig S7(h).

Response/corrections: We thank the Reviewer greatly for these constructive comments. Our responses to each comment are listed as below.

(a) why AA/AB stacking are preferred. Which one is more energetic favorable?

To our best knowledge, there had already been several theoretical calculations on bilayer MoS₂ with at least five different stacking patterns before bilayer MoS₂ was synthesized in experiment. All these works reached the same conclusion that AA stacking with the twist angle of 0° (corresponding to 3R like phase) and AB stacking with the twist angle of 60° (corresponding to 2H phase) are the most energetically stable bilayer crystals [Nano Lett. 2015, 15, 8155; J. Phys. Chem. C 2014, 118, 9203]. This is because the alignments of Mo and S atoms between different layers under such two stacking conditions can make the energy of the system smaller than those under other stacking conditions.

With the development of experimental synthesis technology in recent years, bilayer MoS₂ crystals have been successfully synthesized by CVD and these bilayer crystals typically exhibit only two stacking patterns namely AA and AB stacking corresponding to 3R like and 2H phase, respectively [Nano Lett. 2015, 15, 8155; ACS Nano 2015, 9, 12246; Nanoscale 2017, 9, 13060]. All these reported experimental results collectively suggest that AA and AB stacking patterns are the most thermodynamically stable ones. In a word, AA (3R like phase) and AB (2H phase) stacking phases are the most energetically stable and commonly observed phases in natural and synthetic MoS₂ bilayer crystals [Nanoscale 2017, 9, 13060]. In addition, theoretical calculations [J. Phys. Chem. C 2017, 121, 22559; Phys. Rev. B 2016, 93, 041420] have demonstrated that AA and AB stacking MoS₂ bilayer crystals can be nearly energetically degenerate at room temperature. Free energy calculation shows that AB (2H phase) stacking may become progressively more stable with increasing temperature [Phys. Rev. B 2014, 89, 520], and this is quite consistent with our experimental observations.

We also performed theoretical calculations using DFT-D2 correction method on all five possible high-symmetry stacking orders: (a) AB' (point group D_{3h}): eclipsed stacking with Mo over Mo and S over S; (b) AB (point group D_{3d}): eclipsed stacking with Mo over S, characteristic of the 2H phase; (c) A'B (point group D_{3d}): staggered stacking with S over S; (d) AA (point group C_{3v}): staggered stacking with S over Mo, characteristic of the 3R phase; (e) AA' (point group D_{3d}): staggered stacking with Mo over Mo. The corresponding side and top views of the five stacking orders are shown in Figure R2. It is clear that one can

transform one stacking poly-typism into another by horizontal layer sliding and/or by rotation around the vertical axis. Table R2 provides the detailed comparison in terms of lattice constant (A), Mo-S bond length (L), S-Mo-S bond angle (θ) and total energy (E_{total}) between the five stacking orders. The corresponding bulk experimental values [Z. Anorg. Allg. Chem. 1986, 540, 15] are used as a reference.

Figure R2. Side view and top views of the five possible high-symmetry stacking orders: (a) AB' (point group D3h): eclipsed stacking with Mo over Mo and S over S; (b) AB (point group D3d): eclipsed stacking with Mo over S, characteristic of the 2H phase; (c) A'B (point group D3d): staggered stacking with S over S; (d) AA (point group C3v): staggered stacking with S over Mo, characteristic of the 3R phase; (e) AA' (point group D3d): staggered stacking with Mo over Mo.

Table R2. Detailed Comparison in terms of lattice constant (A), Mo-S bond length (L), S-Mo-S bond angle (θ) and total energy (E_{total}) for the five stacking orders: AB', AB, A'B, AA and AA'. The corresponding bulk experimental values [Z. Anorg. Allg. Chem. 1986, 540, 15] are used as a reference.

Stacking	A (Å)	L (Å)	θ (°)	E_{Total} (eV)
AB'	3.161	2.41	81.76	-4830.5830
AB	3.161	2.41	81.78	-4830.6388
A'B	3.161	2.41	81.75	-4830.0770
AA	3.161	2.41	81.75	-4830.6372
AA'	3.161	2.41	81.77	-4830.6209
Bulk (Expt.)	3.160	2.42	82.00	

As seen, the lattice parameters including lattice constant, bond length and bond angle match well with the experimental ones for all the five stacking orders. Of all the five stacking orders,

AB stacking possesses the smallest system energy ($E_{\text{total}}=-4830.6388$ eV), which is only 1.6 meV lower than that ($E_{\text{total}}=-4830.6372$ eV) of AA stacking, while the other three stacking orders AA', AB' and A'B have much larger system energy with 179, 558 and 5618 meV higher than that of AB stacking, respectively, indicating that AB stacking is the most energetic favourable structure and next is AA stacking. The extremely small difference of system energy between AA and AB stacking is consistent with the previous calculation results [Nano Lett. 2015, 15, 8155; J. Phys. Chem. C 2014, 118, 9203] and can explain the occurrence of these two stacking orders in the natural bulk forms and CVD-grown samples [Nano Lett. 2015, 15, 8155; ACS Nano 2015, 9, 12246; Nanoscale 2017, 9, 13060].

We have emphasized this point in Page 8 of the revised manuscript, which is copied below for the Reviewer's convenience. We have also incorporated this discussion into SI. Please refer to the paragraphs marked by red colour on Pages 6, 7, 14-16 of SI.

The fact that only AA and AB stacking MoS₂ bilayer grains were obtained in experiment means that they are the most stable ones, and this is in good agreement with the previously calculated results^{37, 49} and experimental results^{37, 38, 50}. The detailed mechanism behind this can be found in Supplementary Information using extensive analysis of literature as well as theoretical calculations.

(b) what causes the triangle (hexagonal) shape of the CVD grown MoS₂?

As far as we know, DFT calculation is insufficient to explore the mechanism behind the shape formation of CVD grown MoS₂, and Wulff construction principles or Gibbs-Curie-Wulff equilibrium morphologies for crystal shape selection are needed. Similar calculations have been performed by Cao et al. [J. Phys. Chem. C 2015, 119, 4294] and Schweiger et al. [J. Catal. 2002, 207, 76]. Therefore, we used extensive analysis of the literature to support our experimental findings.

Based on the principles of crystal growth, the shape of a crystal is determined by the growth rate of different crystal faces. The fastest growing faces either become smaller or disappear while the slowest growing faces become the largest. For CVD grown MoS₂, the most commonly observed growing faces are Mo zigzag (Mo-zz) terminations and S zigzag (S-zz) terminations, which can be influenced by the ratio of Mo and S atoms on the growing substrate [Nano Lett. 2015, 12, 8155]. In detail, in a Mo sufficient atmosphere, S-zz terminations grow faster than the Mo-zz terminations, so the domain shape will be a triangle

with three sides of Mo-zz terminations. When the Mo:S ratio is close to the stoichiometric ratio of MoS₂ (1:2), the termination stability and the probability of meeting free atoms are similar for both Mo-zz and S-zz terminations, which results in similar growing rates. In this case, the final shape of the domains will be hexagon. In a S sufficient atmosphere, the domain shape will also transform to triangular with three sides of S-zz terminations. Furthermore, the ratio of Mo and S atoms on the growth substrate may vary along the gas flow direction, leading to the coexistence of different shapes.

In theory, Cao et al. [J. Phys. Chem. C 2015, 119, 4294] revealed that the chemical potential played a crucial role in the determination of equilibrium shape of MoS₂ using DFT calculations and Wulff construction rule. In a S-rich condition, the S chemical potential is high and the Mo edge with S termination (named zz-S2) is the most stable edge structure because its formation energy is lowest. On the basis of crystal growth theory, the low-energy edges will preserve and active edges will disappear during the growth. Therefore, such a S-rich condition will lead to a triangular equilibrium shape of MoS₂ in the end. When the concentration of S is reduced in the growth process, the S chemical potential will decrease. In this situation, the zz-S and zz-S2 edges will be enlarged, finally leading to a hexagonal shape of MoS₂. Schweiger et al. [J. Catal. 2002, 207, 76] have also reached the same conclusion that a high chemical potential of S may result in triangular-shaped particles terminated by the Mo-edge terminated surface based on DFT calculations and Gibbs-Curie-Wulff equilibrium morphologies.

In our case, we used 15 mg MoO₃ powder and 100 mg S powder, which corresponded to a S-rich atmosphere. Therefore, the resultant domains are dominantly triangle as observed. However, as the reaction continued, the S concentration would reduce and vary along the gas flow direction especially on the growing surface inevitably, so the S chemical potential would decrease and the Mo:S ratio on the growth substrate would even be close to 1:2. In such case, hexagon domain shapes would be obtained. Indeed, triangular and hexagonal MoS₂ domains can coexist on the growth substrate as observed.

The relevant clarifications and discussions supported by the extensive analysis of literature can be found on pages 3-5 (marked by red colour) of the Supplementary Information.

(c) if more than bi-layer are grown, which stacking is energetic favorable.

For trilayer, AAA, AAB, ABA and ABA stacking are more energetic favourable. This is because the layer by layer growth is universal for bilayer, trilayer and even multilayers. MoS₂ is interacted by weak van der Waals force between layers and between the bottom layer and the substrate. For trilayer, we can treat the bottom layer as the substrate and the second layer as the first layer as displayed in the previous report [Phys. Rev. B, 2016, 93, 041420]. As such, the stacking order of the third layer with the second layer can be the same as bilayer models. When the bottom bilayer MoS₂ is with AA stacking order, the third layer can has a twist angle of 0° or 60° with the second layer, which can be labelled as AAA or AAB stacking following the nomenclature of bilayer system. When the bottom bilayer MoS₂ is with AB stacking order, the third layer can also has a twist angle of 0° or 60° with the second layer, which can be labelled as ABA or ABB stacking. The trilayer MoS₂ crystals with such four stacking patterns including AAA, AAB, ABA and ABB have indeed been synthesized by our reverse-flow chemical vapour epitaxy method (See Figure S7(a)-(f)), proving that they are energetically favourable configurations, and this is quite consistent with the previous report [Nano Lett., 2015, 15, 8155]. Therefore, we can reach a conclusion that every upper layer can has a twist angle of 0° or 60° with its neighborhood bottom layer for every energetically favourable MoS₂ crystals including bilayer, trilayer, four-layer and even multilayers. As shown in both Fig. 2 and Fig. S7, the twist angles between the upper and bottom layers are always 0° and 60° for bilayer, trilayer, and even multilayer crystals.

Yan et al. [Nano Lett. 2015, 15, 8155] calculated the adhesion energies of four-layer MoS₂ in theory to understand their growth behaviors. They started with a monolayer MoS₂, then added one more layer each time and explored all the possible configurations. As shown in Fig. R3 [Nano Lett. 2015, 15, 8155], they gave the following remarks: at the initial stage of MoS₂ CVD growth, MoS₂ can adopt various stacking configurations due to the comparable adhesion energies. This is the reason why we can obtain a lot of trilayer and multilayer MoS₂ crystals with various stacking patterns on the same growth substrate under certain conditions.

Figure R3 Calculated adhesion energies for few layer MoS₂ growth in a successive manner. The adhesion energy for the nth layer MoS₂ is estimated by subtracting the energy of n-1 layer and the single layer MoS₂, as illustrated in the inset.

The relevant clarifications and discussions supported by the analysis of literature can be found on page 11 (marked by red colour) of the Supplementary Information.

(d) what is the reason behind triangle orientation change in Fig S7(h).

The optical images shown in Fig S7(h) are respectively four-layer MoS₂ crystal with ABAA stacking order (left) and five-layer MoS₂ crystal with AABAB stacking order (right). There are also many multilayer MoS₂ crystals with other stacking patterns as shown in Fig S7(g). As analysed above, when the MoS₂ grow layer by layer, MoS₂ can adopt various stacking configurations due to the comparable adhesion energies at the initial stage of the CVD growth. We have just realized the selectivity between AA and AB stacking for bilayer crystals. The growth parameters for trilayer crystal (including the selectivity between the AAA and AAB stackings as well as between the ABA and ABB stackings) and even multilayer crystal at this stage are not optimised and it will be the subject of future work.

REVIEWERS' COMMENTS:

Reviewer #5 (Remarks to the Author):

No further comments.